# PRAGMATIC CURIOSITY: UNIFYING BAYESIAN OPTIMIZATION AND EXPERIMENTAL DESIGN VIA ACTIVE INFERENCE

## ABSTRACT

Bayesian Optimization (BO) and Bayesian Experimental Design (BED) have traditionally offered separate solutions for goal-oriented and information-oriented tasks, respectively, leaving a gap in complex problems where learning and optimization are not separate phases but deeply intertwined objectives. In this paper, we provide the first unified framework of BO and BED, which is rooted in the principles of active inference (AIF). We introduce "pragmatic curiosity", a new paradigm where the classic explore-exploit dilemma is resolved by minimizing a single objective: the Expected Free Energy (EFE), which naturally balances pragmatic (goal-seeking) and epistemic (information-seeking) drives. We demonstrate the power of this approach on a suite of challenging hybrid tasks, including constrained system identification, targeted active search, and composite optimization with unknown preferences. Empirical results prove the cross-domain adaptability and effectiveness of our proposed framework: our "pragmatic curiosity" paradigm consistently outperforms standard baselines in BO and BED, demonstrating quantifiable improvements in key metrics like estimation accuracy, critical region coverage, and final solution quality.

## 1 INTRODUCTION

Engineering and scientific applications often involve expensive optimization tasks aimed at identifying optimal designs or desired system states. Bayesian optimization (BO) seeks to accelerate this search toward a specified *goal* (Shahriari et al., 2016; Frazier, 2018), while Bayesian experimental design (BED) aims to maximize the *information* gained about unknown system parameters (Rainforth et al., 2023). Both methodologies leverage probabilistic models and acquisition criteria that quantify the utility of evaluating unknown configurations, tailored to either *optimization* or *learning* objectives. Despite their individual successes and the explosion of research in each field, their disconnection creates a vacuum for a broad class of hybrid problems that routinely require simultaneously both *seeking knowledge* and *achieving goals*.

For many real-world applications, such as goal-directed planning (Lookman et al., 2019), environmental monitoring (Konakovic Lukovic et al., 2020), and targeted material design (Matsumoto et al., 2025), *learning* and *optimization* are not separate phases but deeply intertwined objectives. This challenge fundamentally arises across tasks with increasing complexity in terms of both *epistemic consideration* (*i.e.*, from parametric to non-parametric models) and *pragmatic evaluation* (*i.e.*, from known to unknown goals):

**(1) Constrained System Identification**, where the *epistemic* desire of precisely learning a system's parameters is governed by the *pragmatic* need to keep experiments within safe or valid operational bounds (*e.g.*, avoiding sensor saturation, or dangerous chemical reaction). This type of task can be found in numerous applications, including environmental monitoring (Konakovic Lukovic et al., 2020) and catalyst design (Zhong et al., 2020).

**(2) Targeted Active Search**, where the *pragmatic* objective to discover regions that meet certain criteria (*e.g.*, system failure modes, or specific performance ranges) requires *epistemic* curiosity to explore the region's shape, size, and boundaries. Example applications can be found in failure discovery (Ramanagopal et al., 2018) and medical monitoring (Malkomes et al., 2021).

**(3) Composite Bayesian Optimization**, where the *pragmatic* goal is to find an optimal design based on a user's hidden preferences—a task that is impossible without first being *epistemically* curious about the user's objectives themselves. Such scenarios commonly arise in simulation-based design (González & Zavala, 2025; Coelho et al., 2025) and A/B testing (Bakshy et al., 2018)

Classically, to address these hybrid problems, practitioners have been forced to choose between specialized tools, and accommodate problem-specific adaptation by leveraging information-gain criteria to enhance optimization and vice versa. On the BO side, Russo & Van Roy (2018) proposed information-directed sampling (IDS) to online optimization problems. Hvarfner et al. (2023) introduced a statistical distance-based active learning (SAL) criterion into the BO loop to actively learn the Gaussian process hyperparameter even as it searches for the optimum. On the BED (also known as Bayesian active learning, BAL) side, Smith et al. (2023) proposed an expected predictive information gain (EPIG) criterion that focuses on information gain in model predictions, mitigating classical BAL's tendency to select out-of-distribution or low-relevance queries by accounting for an input data distribution. These efforts begin to show the growing synergy between BO and BED techniques, but they remain problem-specific and rarely generalize across categories. More importantly, they lack a principled account of the underlying connections and potential synergy between BO and BED criteria, nor is there a coherent, generalizable framework that unifies them.

In this paper, we present a unified perspective that arises naturally from the principles of Active Inference (AIF) (Friston, 2010; Friston et al., 2017). Originally developed in computational neuroscience, AIF has been successfully applied in fields such as robotics (Lanillos et al., 2021), cognitive science (Smith et al., 2022), and ecosystem modeling (Friston et al., 2024). It prescribes behavior that minimizes expected free energy (EFE)—a quantity that naturally balances an *epistemic* drive for information gain with an *pragmatic* drive to achieve preferred outcomes. We prove that many acquisition strategies in BO and BED can all be interpreted as special cases of EFE minimization.

Building on this unified framework, we propose a new paradigm, conceptualized as "pragmatic curiosity", to offer a principled resolution to the classic explore-exploit dilemma. Under this paradigm, *seeking knowledge* and *achieving goals* are not treated as competing objectives to be balanced, but as two inseparable facets of a single imperative to minimize EFE. As a result, the balance between exploration and exploitation is not manually tuned but emerges from the agent's beliefs, dynamically adapting based on its current uncertainty and the value of potential rewards.

Our "pragmatic curiosity" framework excels at handling the hybrid objectives of *learning* and *optimization* by operating from a single principle. This overcomes a core limitation of pure BO and BED approaches, whose acquisition functions are myopically focused on either exploiting rewards or reducing uncertainty, rendering them ineffective when a task demands a principled combination of both. To validate this, we conduct experiments structured around the three problem classes mentioned above, drawing on applications that include environmental monitoring in plume fields (Konakovic Lukovic et al., 2020), failure detection in autonomous driving scenarios (Ramanagopal et al., 2018), and distributed energy resource allocation in power grids (Kianmehr et al., 2019).

The empirical results reveal a consistent pattern of superior performance, demonstrating our framework's advantages in its ability to solve complex, hybrid objectives. In constrained system identification tasks, our algorithm achieved near-perfect estimation accuracy while requiring up to 40% fewer queries than other methods. For targeted active search tasks, it demonstrated a more effective exploration strategy, discovering a crucial 10% more of the critical failure region within the same budget. Most notably, in tasks with unknown user preferences, our approach always successfully learned the underlying objective where other baselines fail to capture. Together, these findings validate the power of our unified approach, showing that a principled balance between pragmatic and epistemic drives leads to tangible gains across diverse and challenging problem settings.

In summary, our main contributions are the following:

- A formal unification of BO and BED within the AIF framework, overcoming the myopic exploitation of pure BO and the random exploration of pure BED.

- A "pragmatic curiosity" paradigm, which leads to acquisition functions that dynamically arbitrate between goal-seeking and information-seeking based on the single objective of minimizing EFE.

- Comprehensive empirical validation across three categories that differ substantially from one another in terms of both *epistemic consideration* and *pragmatic evaluation*.

## 2 PRELIMINARIES

### 2.1 BAYESIAN OPTIMIZATION

Given an unknown objective function $f : \mathcal{X} \mapsto \Re$, BO seeks to identify the input $x^*$ that maximizes the objective $f$ over an admissible set of queries $\mathcal{X}$, *i.e.*, $x^* = \arg\max_{x \in \mathcal{X}} f(x)$. To achieve this goal, BO relies on a *surrogate model* that provides a probabilistic representation of the objective $f$, and uses this information to compute an *acquisition function* to drive the selection of the most promising sample to query.

**Surrogate model.** We assume the available information regarding the objective function $f$ be stored in the dataset $\mathcal{D}_t := \{(x_1, y_1), \ldots, (x_t, y_t))\}$, where $y_t \sim \mathcal{N}(f(x_t), \sigma^2(x_t))$ is the noisy observation of the objective function by assuming the noise follows a zero-mean normal distribution with a standard deviation $\sigma$. The surrogate model depicts possible explanations of $f$ as $f(x) \sim p(f(x)|\mathcal{D}_t)$ applying a joint distribution over its behavior at each sample $x \in \mathcal{X}$. In Bayesian inference, the prior distribution of the objective $p(f(x))$ is combined with the likelihood function $p(\mathcal{D}_t|f(x))$ to compute the posterior distribution $p(f(x)|\mathcal{D}_t) \propto p(\mathcal{D}_t|f(x))p(f(x))$, representing the updated beliefs about $f(x)$. Typically, Gaussian processes (GPs) have been widely used as the surrogate model for BO due to their efficient posterior sampling that enables cheap, gradient-based optimization of the acquisition function to propose new query points. GP is specified by a joint normal distribution $p(f(x)|\mathcal{D}_t) = \mathcal{N}(\mu_t(x), \kappa_t(x, x'))$ with mean $\mu_t(x)$ and kernel function $\kappa_t(x, x')$, where $\mu_t(x)$ represents the prediction and $\kappa_t(x, x')$ the associated uncertainty.

**Acquisition function.** The surrogate model is utilized to decide the next sample $x_{t+1} \in \mathcal{X}$ through the maximization of an acquisition function $\alpha : \mathcal{X} \mapsto \Re$, *i.e.*, $x_{t+1} = \arg\max_{x \in \mathcal{X}} \alpha(x|\mathcal{D}_t)$, where $\alpha(x|\mathcal{D}_t)$ provides a measure of the improvement that the next query is likely to provide with respect to (w.r.t.) the current surrogate model of the objective function. Many acquisition functions have been proposed, including *probability of improvement* (Močkus, 1975), *expected improvement* (Jones et al., 1998), *upper confidence bound*, and various *entropy search* methods (Hennig & Christian J. Schuler, 2012; Hernández-Lobato et al., 2014; Wang & Jegelka, 2017; Hvarfner et al., 2022a; Neiswanger et al., 2021), as well as practical approaches to optimize them (Wilson et al., 2018).

### 2.2 BAYESIAN EXPERIMENTAL DESIGN

Rather than optimizing an objective function $f(x)$, the purpose of BED is to sequentially select a set of experimental designs $x \in \mathcal{X}$ and gather outcomes $y$, to maximize the amount of information obtained about certain *parameters of interest*, denoted as $\theta$. The parameters $\theta$ can correspond to some explicit model parameters, or any implicitly defined quantity of interest (*e.g.*, the optimum of a function, the output of an algorithm, or future downstream predictions).

Based on the current history of experiments $\mathcal{D}_t := \{(x_1, y_1), \ldots, (x_t, y_t))\}$, BED seeks to find the next experimental design $x_{t+1}$ by maximizing the *expected information gain* (EIG) (Chaloner & Verdinelli, 1995) that a potential experimental outcome $y_{t+1}$ can provide about $\theta$, measured in terms of expected entropy reduction of the posterior distribution of $\theta$:

$$\text{EIG}(x|\mathcal{D}_t) = H[p(\theta|\mathcal{D}_t)] - \mathbb{E}_{p(y|x,\mathcal{D}_t)}[H[p(\theta|\mathcal{D}_t \cup (x, y))]] = I(\theta; (x, y)|\mathcal{D}_t),$$

where $H(\cdot)$ and $I(\cdot)$ denote the entropy and mutual information, respectively.

## 3 A UNIFIED VIEW OF ACQUISITION STRATEGIES

The acquisition strategies in BO typically lead to *goal-directed* behavior, where the (implicit) goal is the optimum of a certain (unknown) objective function. On the contrary, the acquisition strategies in BED encourage *information-seeking* behavior, aiming to gather the maximum amount of information about certain parameters of interest. Although they both can be viewed as realizations of adaptive sampling (Di Fiore et al., 2023), there are no transferable approaches between these two domains due to distinct directives Hvarfner et al. (2025).

In this section, we show that these two seemingly competing imperatives can be naturally balanced through the principles of AIF.

## 3.1 Active Inference as Expected Free Energy Minimization

We specify a probabilistic surrogate model $q(\cdot)$ to capture the relationship between an outcome $y$ and a decision variable $x$ based on a set of parameters $s$ that are of interest, which factorizes as

$$q(x, y, s) \coloneqq p(x, y|s)q(s), \tag{1}$$

where $q(x, y, s)$ is a joint probability distribution over $(x, y, s)$ based on a surrogate distribution $q(s)$ over $s$. We use $q(\cdot)$ to explicitly distinguish this model to be an "surrogate" of the "true" model $p(\cdot)$.

One special aspect of AIF is the way it formalizes goals. Instead of specifying the goal with additional variables related to "rewards" or "costs", AIF directly encodes the preferences over possible outcomes $y$ through a probability distribution $p(y)$. In this distribution, outcomes with higher probabilities are treated as more rewarding. The deviation between observed outcomes and those desired is measured through an information-theoretic quantity known as *self-information* or *surprisal*: $-\log p(y)$. Intuitively, it's a measure of how unexpected an outcome $y$ is, given a prior preference distribution $p(y)$. Consistent with the intuitive notion of surprise, lower probability events generate higher surprisal values. Then the surprisal incurred from an observed outcome $y$ is quantified by

$$\begin{aligned} -\log p(y) = -\log \int p(y, s)ds &= -\log \int \frac{p(y, s)q(s)}{q(s)}ds \\ &= -\log \mathbb{E}_{q(s)}\left[\frac{p(y, s)}{q(s)}\right] \leq -\mathbb{E}_{q(s)}\left[\log \frac{p(y, s)}{q(s)}\right] = F, \end{aligned} \tag{2}$$

where the last inequality follows *Jensen's inequality*, which states that the expectation of a logarithm is always less than or equal to the logarithm of an expectation.

The right-hand side of equation 2 is called the *variational free energy* (VFE), whose name arises from the fact that $F$ resembles the *Helmholtz free energy* in physics. We can see that VFE is always greater than or equal to surprisal (*i.e.*, it is an upper bound on surprisal). In machine learning, the sign of VFE is typically reversed, making it an *evidence lower bound* (ELBO). Maximizing the ELBO is a commonly used optimization approach in machine learning (Titsias, 2009).

To formulate a strategy for decision-making, we need to consider the decision variable $x$ and the outcomes that result from the choice of actions. Since the future outcomes have not yet occurred, we resort to examining the expectation of surprisal over predicted outcomes based on a predictive distribution $q(y|x)$:

$$\begin{aligned} -\mathbb{E}_{q(y|x)} \log p(y|x) \leq -\mathbb{E}_{q(y,s|x)}\left[\log \frac{p(y, s|x)}{q(s|x)}\right] &= -\mathbb{E}_{q(y,s|x)}\left[\log \frac{p(s|x, y)p(y|x)}{q(s|x)}\right] \\ &= -\mathbb{E}_{q(y,s|x)}\left[\log p(s|x, y) - \log q(s|x)\right] - \mathbb{E}_{q(y,s|x)} \log p(y|x) \\ &= -\mathbb{E}_{q(y,s|x)}\left[\log p(s|x, y) - \log q(s|x)\right] - \mathbb{E}_{q(y|x)} \log p(y|x) = G, \end{aligned} \tag{3}$$

where the right-hand side of equation 3 is denoted as the *expected free energy* (EFE).

**Theorem 3.1.** *When using a surrogate model* $q(\cdot) = p(\cdot|\mathcal{D}_t)$ *that is considered as the true model constructed from all available data* $\mathcal{D}_t$, $G$ *can be simplified as*

$$\begin{aligned} G &= -\mathbb{E}_{q(y|x)}\left[D_{KL}(q(s|x, y)||q(s))\right] - \mathbb{E}_{q(y|x)} \log p(y) \\ &= \underbrace{-I_q(s; (x, y))}_{epistemic} \underbrace{-\mathbb{E}_{q(y|x)} \log p(y)}_{pragmatic}, \end{aligned} \tag{4}$$

*where* $I_q(\cdot)$ *represents the mutual information given the surrogate model* $q(\cdot)$.

*Proof.* See Appendix B. □

We can see that by construction, EFE strikes a balance between *information-seeking* and *goal-directed* behavior under some prior preferences. It bounds the difference between *epistemic* value (about parameters) and *pragmatic* value (about outcomes), which captures the imperative to maximize the epistemic value (*i.e.*, information gain about latent states), from interactions with the environment, while maximizing the pragmatic value (*i.e.*, expected preference alignment), regarding

Table 1: A unified view of different acquisition strategies in BO and BED, where $x^*$, $y^*$ represent the true optimal solution and value, respectively, and $\hat{y}$ is the best value observed in $\mathcal{D}_t$.

| Acquisition Strategy | Acquisition Function $\alpha(x)$ | Expected Free Energy | |
|---|---|---|---|
| | | Parameters $s$ | Preferences $p(y)$ |
| Expected Information Gain (Chaloner & Verdinelli, 1995) | $I(\theta; (x,y)\|\mathcal{D}_t)$ | $\theta$ | - |
| Entropy Search (Hennig & Christian J. Schuler, 2012; Hernández-Lobato et al., 2014) | $I(x^*; (x,y)\|\mathcal{D}_t)$ | $x^*$ | - |
| Max-value Entropy Search (Wang & Jegelka, 2017) | $I(y^*; (x,y)\|\mathcal{D}_t)$ | $y^*$ | - |
| Joint Entropy Search (Hvarfner et al., 2022a) | $I((x^*, y^*); (x,y)\|\mathcal{D}_t)$ | $(x^*, y^*)$ | - |
| Bayesian Algorithm Execution (Neiswanger et al., 2021) | $I(\mathcal{O}_\mathcal{A}(f); (x,y)\|\mathcal{D}_t)$ | $\mathcal{O}_\mathcal{A}(f)$ | - |
| GP-Upper Confidence Bound | $\mu_t(x) + \beta^{1/2}\sigma_t(x)$ | $f_\mathcal{X}$ | $\exp\{y\}$ |
| Probability of Improvement (Močkus, 1975) | $p(y \geq \hat{y})$ | - | $\exp\{\mathbb{I}(y \geq \hat{y})\}$ |
| Expected Improvement (Jones et al., 1998) | $\mathbb{E}([y - \hat{y}]_+)$ | - | $\exp\{[y - \hat{y}]_+\}$ |

prior preferences. This crucial aspect of AIF effectively addresses the "explore-exploit dilemma" because the imperatives for exploration and exploitation are just two aspects of EFE:

**Pragmatic Value (Exploitation):** This term encourages goal-directed behavior by favoring actions expected to yield preferred outcomes. Encoded by a prior distribution over desired observations, it functions similarly to a utility or reward function in reinforcement learning (Millidge et al., 2020), driving the agent to exploit its current knowledge to achieve its goals.

**Epistemic Value (Exploration):** This term promotes information-seeking behavior by favoring actions expected to reduce uncertainty about the underlying system maximally. It quantifies the expected information gain about the model's parameters, driving the agent to explore the environment to refine its world model.

## 3.2 REINTERPRETATION OF ACQUISITION STRATEGIES IN BO AND BED

Crucially, minimizing EFE serves as a unifying umbrella principle. Many classic acquisition strategies in BO and BED can be reinterpreted as special cases of minimizing EFE, as shown in Table 1.

The reinterpretations of most of the acquisition strategies in Table 1 are straightforward according to their definitions. However, placing a rather intuitive GP-UCB strategy within this framework seems implicit. To reveal their connection, we rely on the following lemma:

**Lemma 3.2.** *Let $\mathbf{X} \subseteq \mathcal{X}$ be a subset of inputs, and the corresponding function values evaluated at those inputs be denoted as $f_\mathbf{X}$. Given a historical dataset $\mathcal{D}_t$, and new measurements $\mathbf{Y}$ observed at $\mathbf{X}$, the mutual information*

$$I(f_\mathcal{X}; (\mathbf{X}, \mathbf{Y})|\mathcal{D}_t) = I(f_\mathbf{X}; \mathbf{Y}|\mathcal{D}_t),$$

*for any (finite or infinite) set $\mathcal{X}$.*

*Proof.* See Appendix C. $\qquad\square$

Then if we assume constant Gaussian noises $\mathcal{N} \sim (0, \sigma^2)$ for the observations, we have

$$I(f_x; y|\mathcal{D}_t) = H(y|\mathcal{D}_t) - H(y|f_x, \mathcal{D}_t) = \frac{1}{2}\log(1 + \sigma^{-2}\sigma_t^2(x)),$$

where $\sigma_t^2(x)$ is the variance evaluated on the GP model $p(f_x|\mathcal{D}_t)$.

When further assuming that the GP kernel $\kappa_t(x, x') \leq 1, \forall x, x' \in \mathcal{X}$, then $0 \leq \sigma_t^2(x) \leq \kappa_t(x, x') \leq 1$, which gives

$$\log(1 + \sigma^{-2}\sigma_t^2(x)) \geq \log(1 + \sigma^{-2})\sigma_t^2(x).$$

If we choose $\beta = \frac{1}{2}\log(1 + \sigma^{-2})$, then the epistemic term in EFE, *i.e.*, $I(f_x; y|\mathcal{D}_t)$, provides an upper bound of the square of the exploration term $\beta^{1/2}\sigma_t(x)$ in GP-UCB.

This reveals the close relationship between GP-UCB and AIF, showing that even a seemingly pure intuitive strategy, like GP-UCB, can have rather rigorous mathematical foundations underlying it.

## 4 A New Paradigm to Derive Acquisition Functions

The preference distribution $p(y)$ in Table 1 can be interpreted as a softmax transformation of a time-varying value function over the outcomes. Generalizing from this observation, we introduce "pragmatic curiosity", a new paradigm for deriving acquisition functions applicable to a wider class of problems than traditional BO and BED.

Grounded on the *Boltzmann distribution* (also called *Gibbs distribution*) in statistical mechanics, we define an Boltzmann operator $\mathcal{B}_\beta$ that maps an energy function $h(z)$ over domain $\mathcal{Z}$ into a Boltzmann distribution:

$$(\mathcal{B}_\beta[h])(z) \coloneqq \frac{e^{-h(z)/\beta}}{\int_{\mathcal{Z}} e^{-h(z)/\beta}dz}, \tag{5}$$

where $\beta$ is called a temperature parameter (in allusion to statistical mechanics). A higher temperature results in a more uniform output distribution (*i.e.* "more random" with higher entropy), while a lower temperature results in a sharper output distribution, with the maximizers of $h(z)$ dominating.

Then, for any pre-defined energy function $h(y|\mathcal{D}_t)$, we can transfer it into a Boltzmann distribution and use it as the preference distribution $p(y)$ in equation 4:

$$
\begin{aligned}
G &= -I(s; (x, y)|\mathcal{D}_t) - \mathbb{E}_{p(y|x,\mathcal{D}_t)}[\log e^{-h(y|\mathcal{D}_t)/\beta}] + Z \\
&= -I(s; (x, y)|\mathcal{D}_t) + \frac{1}{\beta}\mathbb{E}_{p(y|x,\mathcal{D}_t)}[h(y|\mathcal{D}_t)] + Z,
\end{aligned}
\tag{6}
$$

where $p(y|x, \mathcal{D}_t)$ is the predictive distribution of a surrogate model constructed from the historical data $\mathcal{D}_t$, and $Z = \log \int_{\mathcal{Y}} e^{-h(y|\mathcal{D}_t)/\beta}dy$ is a normalization constant independent of $y$.

Therefore, we propose a new acquisition paradigm by minimizing the EFE, *i.e.*,

$$x_{t+1} = \arg\max_{x \in \mathcal{X}}\{\beta_t I(s; (x, y)|\mathcal{D}_t) - \mathbb{E}_{p(y|x,\mathcal{D}_t)}[h(y|\mathcal{D}_t)]\}, \tag{7}$$

where $\beta_t \geq 0$ is a parameter that modulates the degree of "curiosity", and $h(y|\mathcal{D}_t)$ is a problem-dependent energy function that captures the goal (in terms of regret w.r.t. a certain outcome $y$).

A key innovation of our framework is the problem-dependent energy function, $h(y|\mathcal{D}_t)$, which provides the flexibility to explicitly model uncertainty about the goals themselves. This allows us to derive acquisition functions for a class of complex problems often ignored by standard methods, including tasks with evolving goals (where conditions change over time) or implicit goals (which are not defined a priori). The advantages of this approach on such tasks are demonstrated in the following section.

## 5 Experiments

In this section, we exemplify the benefits and variability of the proposed acquisition strategy by performing experiments across three categories that differ substantially from one another. These problems originate from distinct literature within BO and BED, and none of them are canonical BO or BED tasks. Thus each of them is evaluated against a different set of BO-type (optimization-focused) and BED-type (learning-focused) baselines appropriate for that specific task, ensuring a fair and rigorous comparison.

All acquisitions are evaluated via Monte Carlo estimation using the built-in samplers in BoTorch.

## 5.1 PARAMETRIC MODELS WITH KNOWN INVARIANT GOALS

We first consider a simplest task where the goals are known and invariant, and we have a decent understanding of the model, where only certain finite parameters are unknown.

One typical problem structure is the *constrained system identification* whose objective is to precisely learn unknown system parameters, $\theta$, when valid observations can only be gathered under specific operational constraints, quantified by $C(y) \leq 0$. For instance, when monitoring a chemical plume, the goal is to identify its source ($\theta$) while ensuring that sensor measurements ($y$) do not exceed a saturation threshold ($C(y)$).

In such cases, it is straightforward to choose the interested state as $s = \theta$, and the energy function as $h(y|\mathcal{D}_t) = \mathbb{I}(C(y) > 0)$. This leads to an acquisition function:

$$\alpha(x|\mathcal{D}_t) = \beta I(\theta; (x,y)|\mathcal{D}_t) - \mathbb{E}_{p(y|x,\mathcal{D}_t)}\left[\mathbb{I}(C(y) > 0)\right]. \tag{8}$$

**Tasks.** We perform experiments on a real-world environmental monitoring problem in 2d plume fields where the sensors have a saturation threshold $y_{max}$ (*i.e.*, $C(y) = y - y_{max}$) (Detailed settings and hyper-parameter choices in Appendix D.2). We consider three types of monitoring tasks: (a) locating the unknown source location; (b) estimating unknown wind direction and strength; and (c) identifying the active sources in fields with multiple sources.

**Baselines.** We compare our proposed acquisition strategy (AIF) with BO-type and BED-type baselines tailored this task: (a) Random; (b) Greedy by choosing the point that leads to the least probability of violating the constraint (BO-type); and (c) EIG about the unknown parameter (BED-type).

**Evaluations.** We evaluate the performance from both epistemic and pragmatic perspectives: (a) estimation accuracy; and (b) constraint violations.

**Results.** Fig. 1 shows that our approach achieves consistently stronger query efficiency over baselines while respecting all operational constraints, with cumulative constraint violation always being zero. This advantage is especially evident in the source localization task, where the drive for information and the need to satisfy constraints create opposing pressures. Resolving this conflict, our approach reaches near-perfect estimation using up to 40% fewer queries than competing methods.

**Choices of $\beta$.** To study the effect of $\beta$, we perform an ablation over different $\beta$ values and report the estimation error in Fig. 2 (cumulative constraint violations are zero in all cases). The results show that the optimal $\beta$ is task-dependent. From task (a) to (c), the correlation between sensor measurements and latent parameters weakens, so the mutual information term in equation 8 shrinks. In these less informative regimes, a larger $\beta$ is needed to rescale the information-gain term. However, if $\beta$ is too large, the acquisition becomes overly exploratory, which in turn degrades estimation performance. This yields a simple guideline: start with a moderate $\beta$, increase it only when the mutual information term in equation 8 is consistently small (*e.g.*, due to weak sensor–parameter coupling), and stop or decrease $\beta$ once further increases no longer reduce, or start to increase, the estimation error. In practice, $\beta$ should be just large enough for the curiosity term to matter, but not so large that it induces unnecessary exploration.

## 5.2 NON-PARAMETRIC MODELS WITH KNOWN EVOLVING GOALS

Next, we consider a more challenging setting where the task condition evolves, and the model is fully black-box such that we need to resort to a non-parametric model (*e.g.*, GPs).

One such example is the *targeted active search* in multi-objective design problems. The objectives are viewed as metrics where specific ranges carry particular significance, and the goal is to design experiments that maximize coverage of these important regions $\mathcal{S}$.

Adopted from (Malkomes et al., 2021), we assume that the experimental design problems possess a sense of known resolution, such as simulation accuracy or manufacturing precision/tolerance, and any outcome within distance $\delta$ of another does not convey extra information about $\mathcal{S}$. Similar to (Malkomes et al., 2021), we define the coverage neighborhood of any $y$ as $\mathbb{C}_\delta(y) := \{y' : d(y, y') < \delta\}$. And the coverage neighborhood of a set of points $Y$ is defined as $\mathbb{C}_\delta(Y) := \bigcup_{y \in Y} \mathbb{C}_\delta(y)$.

Since the goal is to cover as much volume of the interested region $\mathcal{S}$ as possible, $h(y|\mathcal{D}_t)$ can be chosen as $h(y|\mathcal{D}_t) = \text{Vol}(\mathcal{S}) - \text{Vol}(\mathbb{C}_\delta(Y \cup y) \cap \mathcal{S})$, where $Y$ contains all the history of outcomes.

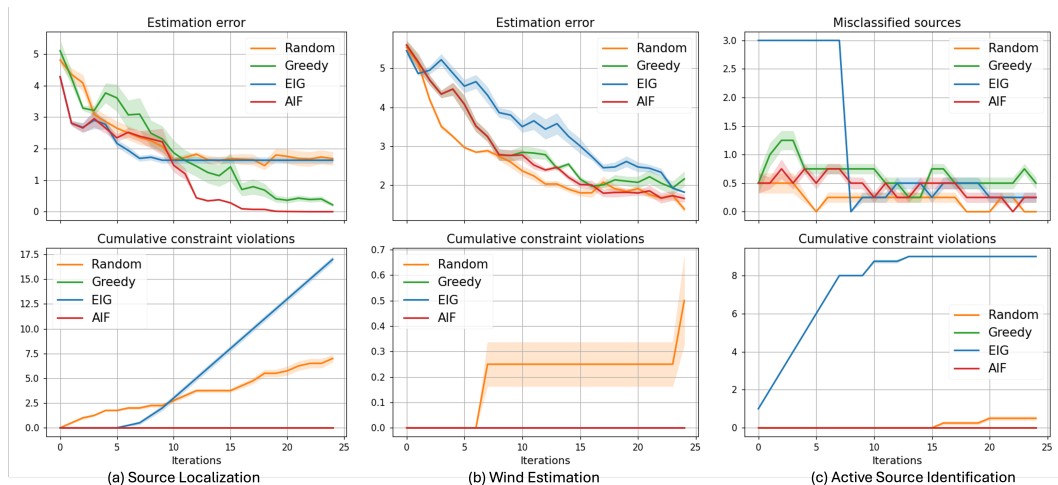

Figure 1: Performance evaluation for constrained system identification on environmental monitoring in 2d plume fields. Error bars represent $\pm 1$ std over 5 seeds.

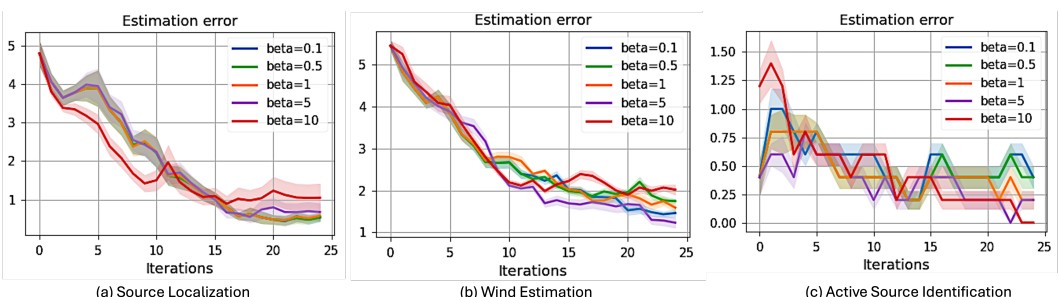

Figure 2: Ablation studies of $\beta$ for constrained system identification on environmental monitoring in 2d plume fields. Error bars represent $\pm 1$ std over 5 seeds.

Similarly, coverage over the design configurations can be flexibly incorporated by expanding the interested state into the whole input space, *i.e.*, $s = f_{\mathcal{X}}$. With non-parametric models like GPs, we can leverage Lemma 3.2 and derive an acquisition function:

$$\alpha(x|\mathcal{D}_t) = \beta I(f_x; y|\mathcal{D}_t) + \mathbb{E}_{p(y|x,\mathcal{D}_t)}[\text{Vol}(\mathbb{C}_\delta(Y \cup y) \cap \mathcal{S})]. \tag{9}$$

**Tasks.** We perform experiments on a real-world failure discovery problem in autonomous driving scenarios where the perception module, a YOLO detector, may fail due to multiple causes, which could potentially lead to collisions (3d input-2d output). We consider three target sets with decreasing volume, *i.e.*, Target Set 1 $\supset$ Target Set 2 $\supset$ Target Set 3 (Detailed settings and hyper-parameter choices in Appendix D.3).

**Baselines.** We again compare our proposed acquisition strategy (AIF) with BO-type and BED-type baselines tailored this task: (a) Random; (b) Greedy by maximizing coverage volume in metrics space (BO-type); and (c) EIG by maximizing coverage volume in parameter space (BED-type).

**Evaluations.** We evaluate the performance by inspecting the coverage volume over both spaces: (a) metrics space ($\mathbb{C}_\delta(Y)$); and (b) parameter space ($\mathbb{C}_\delta(X)$).

**Results.** As shown in Fig. 3, our AIF algorithm effectively balances coverage of both the parameter and metrics spaces. This capability is especially impactful for smaller target sets where the search is more difficult. In the most challenging case (Target Set 3), our method identifies nearly 10% more of the critical failure region compared to the leading baseline.

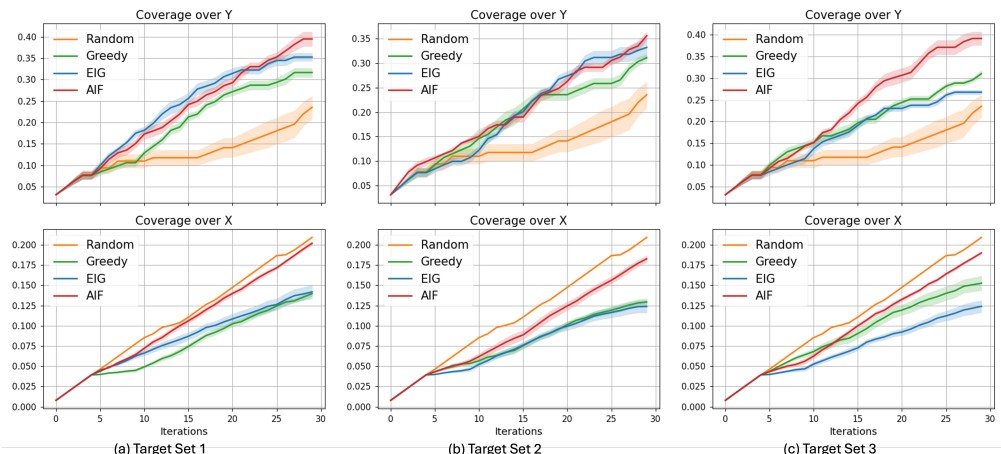

Figure 3: Performance evaluation for targeted active search on failure discovery in autonomous driving scenarios. Error bars represent $\pm 1$ std over 4 seeds.

### 5.3 NON-PARAMETRIC MODELS WITH UNKNOWN GOALS

Finally, we look into the most difficult setting where both the models and goals are black-box.

One practical scenario arises from the *composite BO* in multi-objective optimization problems. The objectives are weighted by a preference function $g(y)$ that is unknown *a priori* and must be simultaneously learned during the optimization process.

As assumed in (Lin et al., 2022), we can estimate the unknown preference function $g(y)$ by asking a decision-maker (DM) to express preferences over pairs of outcomes $(y_1, y_2)$. Let $z(y_1, y_2) \in \{1, 2\}$ indicate whether the DM preferred the first or second outcome offered. Following (Chu & Ghahramani, 2005), we assume that the DM's responses are distributed according to a *probit* likelihood: $L(z(y_1, y_2) = 1|g(y_1), g(y_2)) = \Phi(\frac{g(y_1)-g(y_2)}{\sqrt{2}\lambda})$, where $\lambda$ is a hyperparameter, and $\Phi$ is the standard normal CDF.

Taking the joint exploration and exploitation of the preference function $g(y)$ into consideration, we can choose $h(y|\mathcal{D}_t)$ as the EFE of the preference model, *i.e.*, $h(y|\mathcal{D}_t) = \text{EFE}(g(y))$, which naturally extends the acquisition function into a nested structure:

$$\alpha(x|\mathcal{D}_t) = \gamma I(f_x; y|\mathcal{D}_t) + \mathbb{E}_{p(y|x,\mathcal{D}_t)}[\beta I(g_y; z|\mathcal{D}_t) + \mathbb{E}_{p(g_y|y,\mathcal{D}_t)}g_y]], \tag{10}$$

where $x = [x_1, x_2]$ are two jointly evaluated candidates, and $\gamma, \beta \geq 0$.

**Tasks.** We perform experiments on three real-world problems, including vehicle safety (5d-3d), penicillin production simulator (7d-3d), and distributed energy resource allocation in power grids (40d-4d) (Detailed settings and hyper-parameter choices in Appendix D.4).

**Baselines.** Different from (Lin et al., 2022), which separates and iterates the *preference exploration* and *experimentation* stages, our approach unifies the exploration and exploitation of both outcome and preference models simultaneously. In its best setting (*i.e.*, switch stages and update models at each step), their acquisition strategy reduces to a special case of our proposed equation 10 with $\beta = \gamma = 0$. Thus, we design a set of baselines to investigate the effect that each component in equation 10 plays for the joint exploration and exploitation of hierarchical models. We ablate the full AIF strategy (g + g IG + f IG) by (a) removing the last term (EIG); (b) removing the first term (g + g IG); and (c) removing the first two terms (g only).

**Evaluations.** We evaluate their performance by inspecting the best preference of all collected outcomes using the true preference function $g(y)$.

**Results.** Fig. 4 illustrates our AIF approach's outstanding ability to learn unknown preference functions, a key advantage over baselines that often fail due to ill-directed queries. As the tasks become more complex and noisy (from (a) to (c)), our results demonstrate that every component of our ac-

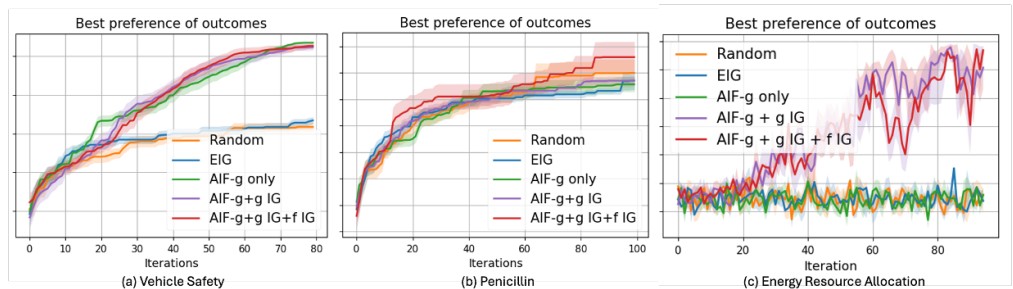

Figure 4: Performance evaluation for composite BO with unknown preferences. Error bars represent $\pm 1$ std over 20 seeds for vehicle safety, penicillin, and 5 for energy resource allocation.

quisition function (10) plays an irreplaceable role in achieving optimal performance. This benefit is most stark in the energy resource allocation task, where competing methods failed to capture any meaningful preference model while our approach consistently succeeded.

**Choices of $\beta$ and $\gamma$.** We also conducted an ablation study to investigate how the curiosity weights $\beta$ and $\gamma$ affect the learning dynamics. Detailed plots and analysis are included in Appendix D.4.3. Based on those results, we suggested the following design guideline. When the preference model $g(y)$ is complex or weakly informed by the initial data, $\beta$ should be set relatively large so that the algorithm explicitly allocates queries to reduce uncertainty in $g$. At the same time, $\gamma$ should be kept in a moderate-to-large range to ensure sufficient exploration of the outcome model $f(x)$. In simpler or more informative tasks, smaller values of $\beta$ and $\gamma$ already suffice, and the method is relatively insensitive to their precise tuning.

**Benefits of joint learning and optimization.** To highlight the benefits of jointly learning and optimizing, rather than separating these into stages, we compare our method against several BOPE variants from Lin et al. (2022) that use different stage-wise design choices. Detailed experimental setups, design choices, plots, and analyses are provided in Appendix D.4.4. The results show that our method naturally balances exploration and exploitation at each step and consistently discovers higher-preference regions, whereas the BOPE variants are highly sensitive to how the stages are configured. Consequently, stage-wise approaches like BOPE require careful manual tuning of these choices, while our unified formulation automates this trade-off and is therefore more amenable to higher-order hierarchical models.

## 6 CONCLUSION AND LIMITATION

In this work, we unified BO and BED by intducing "pragmatic curiosity", a paradigm grounded in AIF. Our framework resolves the classic explore-exploit challenge by treating goal-seeking and information-seeking as two facets of a single, principled objective: minimizing EFE. We validated this approach across a suite of challenging hybrid tasks, demonstrating that our unified approach consistently outperforms standard baselines.

The potential impact of this work is to promote a conceptual shift from handling the explore–exploit dilemma with ad hoc heuristics to treating it as a unified inference problem. Our contribution is a concrete, principled step in this direction: we introduce an AIF-based acquisition that jointly accounts for learning and optimization, and demonstrate its benefits on tasks where these two objectives are tightly intertwined. We view this as a foundational rather than final step toward unifying *goal-directed* and *information-seeking* objectives, and hope it will invite more attention and follow-up work on this perspective.

The main limitation of our framework is the computational cost of estimating the EFE, which can be prohibitive in high-dimensional or time-critical applications. This points to clear avenues for future research, in particular developing more scalable and efficient approximations of the EFE so that the proposed framework can be applied to larger and more complex real-world problems.

## ETHICS STATEMENT

This research is purely computational and does not involve human subjects, animal testing, or personally identifiable data. The primary ethical considerations relate to the potential downstream applications of our proposed "pragmatic curiosity" framework. We have focused on applications with clear societal benefits: enhancing environmental monitoring, improving safety in autonomous systems, and optimizing energy grid stability.

## REPRODUCIBILITY STATEMENT

We are committed to the full reproducibility of our research.

**Code:** The complete source code for our framework, all baseline algorithms, and the experimental testbeds will be made publicly available in a GitHub repository upon publication of this manuscript. The code is implemented in Python and primarily relies on standard libraries such as PyTorch, GPyTorch, and BoTorch.

**Experiments:** All experimental results reported in this paper are averaged over multiple random seeds to ensure statistical robustness. The exact experiment environments, configurations, and hyper-parameters used to generate every figure and table are provided in Appendix D, allowing for the direct replication of our findings.

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

## A  RELATED WORK

Several independent attempts have been made to handle the hybrid objectives of goal seeking and learning. Below, we discuss some key prior works, originating from three viewpoints: (1) BO based settings, (2) BED (also known as Bayesian active learning, BAL) based works, and (3) some works that have acknowledged the synergy and attempted to present a unifying viewpoint, using adaptive sampling and AIF based approaches.

On the BO side, Russo & Van Roy (2018) proposed information-directed sampling (IDS) to online optimization problems by minimizing the ratio between squared expected single-period regret and a measure of information gain. Hvarfner et al. (2023) introduced a statistical distance-based active learning (SAL) criterion into the BO loop to actively learn the Gaussian process hyperparameter even as it searches for the optimum. Tu et al. (2022); Hvarfner et al. (2022b) propose information-theoretic acquisition function for multi-objective BO, which considers the joint information gain for the optimal set of inputs and outputs. Overall, these works acknowledge and demonstrate the value of information gain in enhancing the capabilities of goal-seeking strategies, with an ultimate focus on optimization. However, they do not explicitly comment on the synergy of goal seeking and learning, or formalize the acquisition strategies as a way to explicitly balance both objectives.

On the BED (also known as Bayesian active learning, BAL) side, Smith et al. (2023) proposed an expected predictive information gain (EPIG) criterion that focuses on information gain in model predictions, mitigating classical BAL's tendency to select out-of-distribution or low-relevance queries by accounting for an input data distribution. Lookman et al. (2019) discusses the performance of various BAL and BO strategies for material design application, noting their connection with adaptive sampling, and simultaneous exploration and exploitation. This shows that realistic application of BO/BED based approaches often necessitates a formalized viewpoint unifying each distinct objectives.

These efforts begin to show the growing synergy between BO and BED techniques, where information-gain criteria are leveraged to enhance optimization and vice versa. In search of a unifying perspective for these techniques, Di Fiore et al. (2023) views BO and BAL as symbiotic adaptive sampling techniques driven by common principles of 'goal-driven learning'. However, their viewpoint is limited to BO and BED as separate realizations of adaptive sampling, and they do not discuss a way to combine both objectives in a single acquisition strategy.

In this paper, we present a unified perspective that arises naturally from the principles of Active Inference (AIF) (Friston, 2010; Friston et al., 2017). Originally developed in computational neuroscience, AIF has been successfully applied in fields such as robotics (Lanillos et al., 2021), cognitive science (Smith et al., 2022), ecosystem modeling (Friston et al., 2024), and goal-directed planning in partially observable environments Matsumoto et al. (2025). These works leverage the ability of AIF to accommodate goal-directed and epistemic considerations, which we utilize as the backbone for our unifying mechanism discussed in this paper.

## B  PROOF OF THEOREM 3.1

*Proof.* Recall that

$$G = \underbrace{-\mathbb{E}_{q(y,s|x)}\left[\log p(s|x,y) - \log q(s|x)\right]}_{\text{Term 1}}\underbrace{-\mathbb{E}_{q(y|x)}\log p(y|x)}_{\text{Term 2}}$$

Since our purpose is to align the true outcome distribution given the decision $x$ (*i.e.*, $p(y|x)$) with the prior preference distribution $p(y)$, we replace the second term of $G$ by $-\mathbb{E}_{q(y|x)}\log p(y)$. It is commonly referred to as *pragmatic* value, which scores the anticipated surprisal of the future outcomes given a predictive distribution $q(y|x)$.

For the first term of $G$, add and subtract $\log q(s|x,y)$ inside the expectation, we get

$$\text{Term 1} = -\mathbb{E}_{q(y,s|x)}\left[\log p(s|x,y) - \log q(s|x,y) + \log q(s|x,y) - \log q(s|x)\right]$$

$$= -\mathbb{E}_{q(y|x)}\mathbb{E}_{q(s|x,y)}\left[\log p(s|x,y) - \log q(s|x,y) + \log q(s|x,y) - \log q(s|x)\right]$$

$$= -\mathbb{E}_{q(y|x)}\left[\mathbb{E}_{q(s|x,y)}\left[\log p(s|x,y) - \log q(s|x,y)\right] + \mathbb{E}_{q(s|x,y)}\left[\log q(s|x,y) - \log q(s|x)\right]\right]$$

$$= -\mathbb{E}_{q(y|x)}\left[\underbrace{-D_{\text{KL}}(q(s|x,y)||p(s|x,y))}_{\text{Term 3}} + \underbrace{D_{\text{KL}}(q(s|x,y)||q(s|x))}_{\text{Term 4}}\right].$$

$$(11)$$

Since the surrogate model $q(s)$ usually will not be updated until a new outcome $y$ is observed, it doesn't depend on the decision variable $x$ solely, *i.e.*, $q(s|x) = q(s)$. Thus Term 4 becomes $D_{\text{KL}}(q(s|x,y)||q(s))$, which is commonly referred to as the *epistemic* value, or the expected information gain of a state when it is conditioned on expected observations.

Term 3 is, in general, intractable since it depends on the true posterior distribution $p(s|x,y)$ that is often unknown. The best surrogate model we can have is $q(\cdot) = p(\cdot|\mathcal{D}_t)$ that is constructed from all available data $\mathcal{D}_t$. Thus if we treat this surrogate model $q(s)$ as the best approximation of the true model $p(s)$, *i.e.*, $q(s) \approx p(s)$, and the surrogate and true models both follow the Bayes' rule, *i.e.*, $q(s|x,y) = \frac{p(x,y|s)q(s)}{\int p(x,y|s)q(s)ds}$, $\quad p(s|x,y) = \frac{p(x,y|s)p(s)}{\int p(x,y|s)p(s)ds}$, then Term 3 vanishes. $\qquad\square$

## C    PROOF OF LEMMA 3.2

To prove Lemma 3.2, we first introduce a lemma:

**Lemma C.1.** *For any (finite or infinite) set $\mathcal{X}$, after a few new measurements $(\mathbf{X}, \mathbf{Y}), \mathbf{X} \subseteq \mathcal{X}$ the KL divergence between $p(f_{\mathcal{X}}|\mathcal{D} \cup (\mathbf{X}, \mathbf{Y}))$ and $p(f_{\mathcal{X}}|\mathcal{D})$ is*

$$D_{KL}[p(f_{\mathcal{X}}|\mathcal{D} \cup (\mathbf{X}, \mathbf{Y}))||p(f_{\mathcal{X}}|\mathcal{D})] = \mathbb{E}_{p(f_{\mathbf{X}}|\mathcal{D})}\left[\frac{L(\mathbf{Y}|f_{\mathbf{X}})}{L(\mathbf{Y})}\log\frac{L(\mathbf{Y}|f_{\mathbf{X}})}{L(\mathbf{Y})}\right],$$

*where $L(\mathbf{Y}|f_{\mathbf{X}})$ is the likelihood of observations, $L(\mathbf{Y}) = \int L(\mathbf{Y}|f_{\mathbf{X}})p(f_{\mathbf{X}}|\mathcal{D})df_{\mathbf{X}}$ the marginal likelihood.*

*Proof.* We first consider the case where $\mathcal{X}$ is *finite* to provide an intuitive insight. The KL divergence between $p(f_{\mathcal{X}}|\mathcal{D} \cup (\mathbf{X}, \mathbf{Y}))$ and $p(f_{\mathcal{X}}|\mathcal{D})$ can be given as

$$D_{\text{KL}}[p(f_{\mathcal{X}}|\mathcal{D} \cup (\mathbf{X}, \mathbf{Y}))||p(f_{\mathcal{X}}|\mathcal{D})]$$

$$= D_{\text{KL}}[p(f_{\mathcal{X}\backslash\mathbf{X}}, f_{\mathbf{X}}|\mathcal{D} \cup (\mathbf{X}, \mathbf{Y}))||p(f_{\mathcal{X}\backslash\mathbf{X}}, f_{\mathbf{X}}|\mathcal{D})]$$

$$= \int p(f_{\mathcal{X}\backslash\mathbf{X}}, f_{\mathbf{X}}|\mathcal{D} \cup (\mathbf{X}, \mathbf{Y}))\log\frac{p(f_{\mathcal{X}\backslash\mathbf{X}}, f_{\mathbf{X}}|\mathcal{D} \cup (\mathbf{X}, \mathbf{Y}))}{p(f_{\mathcal{X}\backslash\mathbf{X}}, f_{\mathbf{X}}|\mathcal{D})}df_{\mathcal{X}\backslash\mathbf{X}}df_{\mathbf{X}}$$

$$= \int p(f_{\mathcal{X}\backslash\mathbf{X}}, f_{\mathbf{X}}|\mathcal{D} \cup (\mathbf{X}, \mathbf{Y}))\log\frac{p(f_{\mathcal{X}\backslash\mathbf{X}}, f_{\mathbf{X}}|\mathcal{D})L(\mathbf{Y}|f_{\mathbf{X}})}{p(f_{\mathcal{X}\backslash\mathbf{X}}, f_{\mathbf{X}}|\mathcal{D})L(\mathbf{Y})}df_{\mathcal{X}\backslash\mathbf{X}}df_{\mathbf{X}}$$

$$= \int p(f_{\mathcal{X}\backslash\mathbf{X}}, f_{\mathbf{X}}|\mathcal{D} \cup (\mathbf{X}, \mathbf{Y}))\log\frac{L(\mathbf{Y}|f_{\mathbf{X}})}{L(\mathbf{Y})}df_{\mathcal{X}\backslash\mathbf{X}}df_{\mathbf{X}} \qquad (12)$$

$$= \int p(f_{\mathbf{X}}|\mathcal{D} \cup (\mathbf{X}, \mathbf{Y}))\log\frac{L(\mathbf{Y}|f_{\mathbf{X}})}{L(\mathbf{Y})}df_{\mathbf{X}}$$

$$= \int \frac{p(f_{\mathbf{X}}|\mathcal{D})L(\mathbf{Y}|f_{\mathbf{X}})}{L(\mathbf{Y})}\log\frac{L(\mathbf{Y}|f_{\mathbf{X}})}{L(\mathbf{Y})}df_{\mathbf{X}}$$

$$= \mathbb{E}_{p(f_{\mathbf{X}}|\mathcal{D})}\left[\frac{L(\mathbf{Y}|f_{\mathbf{X}})}{L(\mathbf{Y})}\log\frac{L(\mathbf{Y}|f_{\mathbf{X}})}{L(\mathbf{Y})}\right],$$

where $L(\mathbf{Y}|f_{\mathbf{X}})$ is the likelihood of observations, $L(\mathbf{Y}) = \int L(\mathbf{Y}|f_{\mathbf{X}})p(f_{\mathbf{X}}|\mathcal{D})df_{\mathbf{X}}$ the marginal likelihood.

Now we move to the more general cases where $\mathcal{X}$ is *infinite*. In such cases, there is no useful infinite-dimensional Lebesgue measure with respect to an "infinite-dimensional vector" $f_{\mathcal{X}}$. Thus, we need to resort to a more general definition for KL divergence based on the Radon-Nikodym derivative:

**Definition C.2.** If $P$ and $Q$ are probability measures over a set $\mathcal{X}$, and $P$ is absolutely continuous with respect to $Q$, then the KL divergence from $P$ to $Q$ is defined as

$$D_{\text{KL}}[P||Q] = \int_{\mathcal{X}} \log(\frac{dP}{dQ})dP,$$

where $\frac{dP}{dQ}$ is the Radon–Nikodym derivative of $P$ with respect to $Q$, and provided the expression on the right-hand side exists.

According to the measure-theoretic definition of Bayes' theorem for a dominated model (Schervish, 1995), the Radon-Nikodym derivative of the posterior $P(\cdot) := p(\cdot|\mathcal{D} \cup (\mathbf{X}, \mathbf{Y}))$ with respect to the prior $\hat{P}(\cdot) := p(\cdot|\mathcal{D})$ is given as

$$\frac{dP}{d\hat{P}}(f_{\mathcal{X}}) = \frac{L(\mathbf{Y}|f_{\mathcal{X}})}{L(\mathbf{Y})}.$$

Since the dataset $\mathbf{Y}$ is finite, so similar to previous, we restrict the likelihood to only depend on the finite dataset:

$$\frac{dP}{d\hat{P}}(f_{\mathcal{X}}) = \frac{L(\mathbf{Y}|f_{\mathbf{X}})}{L(\mathbf{Y})}.$$

Now the KL divergence between $\hat{P}$ and $P$ is quantified as

$$
\begin{aligned}
&D_{\text{KL}}[P(f_{\mathcal{X}})||\hat{P}(f_{\mathcal{X}})] \\
&= \int_{f_{\mathcal{X}}} \log(\frac{dP}{d\hat{P}}(f_{\mathcal{X}}))dP(f_{\mathcal{X}}) \\
&= \int_{f_{\mathcal{X}}} \log(\frac{L(\mathbf{Y}|f_{\mathbf{X}})}{L(\mathbf{Y})})dP(f_{\mathcal{X}}) \\
&= \int_{f_{\mathcal{X}}} \log(\frac{L(\mathbf{Y}|f_{\mathbf{X}})}{L(\mathbf{Y})})\frac{L(\mathbf{Y}|f_{\mathbf{X}})}{L(\mathbf{Y})}d\hat{P}(f_{\mathcal{X}}) \\
&= \int_{f_{\mathbf{X}}} \log(\frac{L(\mathbf{Y}|f_{\mathbf{X}})}{L(\mathbf{Y})})\frac{L(\mathbf{Y}|f_{\mathbf{X}})}{L(\mathbf{Y})}d\hat{P}(f_{\mathbf{X}}) \\
&= \mathbb{E}_{\hat{P}(f_{\mathbf{X}})}[\frac{L(\mathbf{Y}|f_{\mathbf{X}})}{L(\mathbf{Y})} \log \frac{L(\mathbf{Y}|f_{\mathbf{X}})}{L(\mathbf{Y})}],
\end{aligned}
\tag{13}
$$

which has the exact same form as equation 12.

Therefore, we can conclude that regardless of the set $\mathcal{X}$ being finite or infinite, the KL divergence between the prior and posterior only depends on the evaluations of the observed data. That is to say, whilst we are in fact quantifying the KL divergence between the full distributions, we only need to keep track of the distribution over finite function values $f_{\mathbf{X}}$. $\qquad\square$

Now we are ready to prove Lemma 3.2.

*Proof.* According to Lemma C.1,

$$
\begin{aligned}
I(f_{\mathcal{X}};(\mathbf{X},\mathbf{Y})|\mathcal{D}) =& \mathbb{E}_{p(\mathbf{Y}|\mathbf{X},\mathcal{D})}[D_{\mathrm{KL}}[p(f_{\mathcal{X}}|\mathcal{D}\cup(\mathbf{X},\mathbf{Y}))||p(f_{\mathcal{X}}|\mathcal{D})]] \\
=& \mathbb{E}_{p(\mathbf{Y}|\mathbf{X},\mathcal{D})}\mathbb{E}_{p(f_{\mathbf{X}}|\mathcal{D})}\big[\frac{L(\mathbf{Y}|f_{\mathbf{X}})}{L(\mathbf{Y})}\log\frac{L(\mathbf{Y}|f_{\mathbf{X}})}{L(\mathbf{Y})}\big] \\
=& \mathbb{E}_{L(\mathbf{Y})}\mathbb{E}_{p(f_{\mathbf{X}}|\mathcal{D})}\big[\frac{L(\mathbf{Y}|f_{\mathbf{X}})}{L(\mathbf{Y})}\log\frac{L(\mathbf{Y}|f_{\mathbf{X}})}{L(\mathbf{Y})}\big] \\
=& \mathbb{E}_{p(f_{\mathbf{X}}|\mathcal{D})}\mathbb{E}_{L(\mathbf{Y})}\big[\frac{L(\mathbf{Y}|f_{\mathbf{X}})}{L(\mathbf{Y})}\log\frac{L(\mathbf{Y}|f_{\mathbf{X}})}{L(\mathbf{Y})}\big] \\
=& \mathbb{E}_{p(f_{\mathbf{X}}|\mathcal{D})}\int\big[\frac{L(\mathbf{Y}|f_{\mathbf{X}})}{L(\mathbf{Y})}\log\frac{L(\mathbf{Y}|f_{\mathbf{X}})}{L(\mathbf{Y})}\big]L(\mathbf{Y})d\mathbf{Y} \\
=& \mathbb{E}_{p(f_{\mathbf{X}}|\mathcal{D})}\int L(\mathbf{Y}|f_{\mathbf{X}})\log\frac{L(\mathbf{Y}|f_{\mathbf{X}})}{L(\mathbf{Y})}d\mathbf{Y} \\
=& \mathbb{E}_{p(f_{\mathbf{X}}|\mathcal{D})}[D_{\mathrm{KL}}[L(\mathbf{Y}|f_{\mathbf{X}})||L(\mathbf{Y})]] \\
=& I(f_{\mathbf{X}};\mathbf{Y}|\mathcal{D}).
\end{aligned}
\tag{14}
$$

$\square$

# D EXPERIMENTAL DETAILS

This appendix provides a comprehensive overview of the simulation environment, model parameters, and hyper-parameter choices used to generate the results in this paper. The experiments were designed to be reproducible given the configurations outlined below.

## D.1 SIMULATION ENVIRONMENT

All simulations for the perception failure evaluation in CARLA (Section D.3) were conducted on a Linux workstation with Ubuntu 22.04 LTS equipped with an Intel 13th Gen Core i7-13700KF CPU (16 cores, 24 threads, up to 5.4 GHz) and an NVIDIA GeForce RTX 4090 GPU (24 GB VRAM). The system ran CARLA simulations using CUDA 12.2 and NVIDIA driver version 535.230.02.

All other experiments were run on a MacBook Pro equipped with an Apple M2 Pro processor (10-core CPU, 16-core GPU) and a 3024 × 1964 Retina display. The GPU supports Metal 3, and the system was used as-is without external accelerators.

All experiments were conducted using Python 3.9. The core scientific computing libraries utilized were:

- BoTorch (Balandat et al., 2020)
- GPyTorch (Gardner et al., 2018)

In all experiments, we utilized the built-in Monte Carlo sampler in Botorch for the optimization of acquisition functions. The Monte Carlo samples are drawn from the posteriors for each model to approximate the expectations of acquisition functions.

## D.2 CONSTRAINED SYSTEM IDENTIFICATION

### D.2.1 PLUME FIELD MODEL AND PARAMETERS

We consider the monitoring of a chemical plume field, where multiple plume sources generate plume particles that can be measured by sensors. The field function is represented by the rate of hits, defined as the average number of particles per unit time measured by the sensor at a certain location.

The rate of hits for a chemical plume source is given as:

$$
R_\theta(x) = \frac{R_s}{\log\frac{\gamma}{a}}\exp(-\frac{\langle\theta-x,V\rangle}{2D})K_0(\frac{||\theta-x||_2}{\gamma}),
$$

where $\theta$ is the location of the plume source, $R_s$ is the rate at which the plume source releases the plume particles in the environment, $\gamma = \sqrt{D\tau/(1 + \frac{||V||^2\tau}{4D})}$ is the average distance traveled by a plume particle in its lifetime, $a$ is the size of the sensor detecting plume particles, $V$ is the average wind velocity, $D$ is the diffusivity of the plume particles, and $K_0$ is the Bessel function of zeroth order.

The measurement $y$, *i.e.*, the number of particles measured, is modeled as a Poisson random variable with $R_\theta(x)\Delta t$ as the rate parameter, which leads to a likelihood model as

$$L_\theta(y|x) = \frac{\exp(-R_\theta(x)\Delta t)(R_\theta(x)\Delta t)^y}{y!},$$

where $\Delta t$ is the time taken to obtain a measurement.

**General Field and Sensor Parameters:**

- Grid Size: $100 \times 100$ units.
- Sensor Size ($a$): 1.0 units.
- Measurement Time ($\Delta t$): 1.0 seconds.

**Source Parameters:** Eight potential plume sources were defined for the experiments. Their specific physical parameters are detailed as:

| SOURCE | $\theta$ | $R_s$ | $\gamma$ | $V$ | $D$ |
|---|---|---|---|---|---|
| 1 | [20.0, 20.0] | 100.0 | 50.0 | [0.5, 0.5] | 10 |
| 2 | [30.0, 80.0] | 100.0 | 60.0 | [-0.3, 0.2] | 15 |
| 3 | [45.0, 55.0] | 15.0 | 50.0 | [0.5, 0.5] | 10 |
| 4 | [50.0, 50.0] | 18.0 | 30.0 | [-0.3, 0.2] | 15 |
| 5 | [55.0, 45.0] | 16.0 | 40.0 | [0.2, -0.4] | 12 |
| 6 | [48.0, 52.0] | 17.0 | 35.0 | [0.1, 0.1] | 11 |
| 7 | [52.0, 55.0] | 14.0 | 45.0 | [-0.1, -0.1] | 13 |
| 8 | [52.0, 52.0] | 18.0 | 40.0 | [0.1, -0.1] | 13 |

### D.2.2 TASK-SPECIFIC CONFIGURATIONS

**Source Localization.**

- Goal: Estimate the unknown location $\theta = [x, y]$ of a single active source.
- Ground Truth: Source 1 was configured as the single active source.
- Maximum Sensor Threshold ($y_{max}$): 60.0 hits/second.
- Hypothesis Space: A discrete grid of potential locations spanning [0,100]×[0,100] with a resolution of 5 units, resulting in a $20 \times 20$ grid of 400 hypotheses.
- Degree of curiosity: $\beta = 0.5$.

**Wind Estimation.**

- Goal: Estimate the unknown wind vector $V = [v_x, v_y]$ for a single source with a known location.
- Ground Truth: Source 2 was used, with its true wind vector set to [-0.3, 0.2].
- Maximum Sensor Threshold ($y_{max}$): 60.0 hits/second.
- Hypothesis Space: A discrete grid of potential wind vectors spanning [-1,1]×[-1,1] with a resolution of 0.1 units, resulting in a $20 \times 20$ grid of 400 hypotheses.
- Degree of curiosity: $\beta = 1.0$.

**Active Source Identification.**

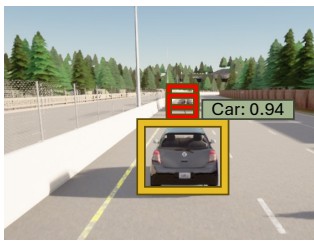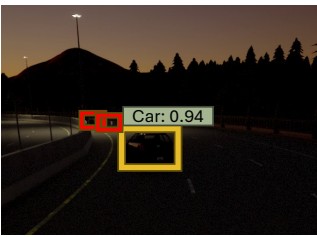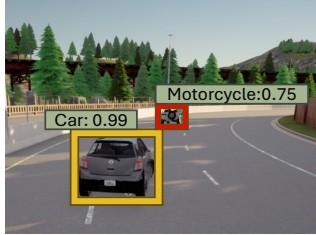

Figure 5: Examples of missed object detection by YOLO due to two reasons considered in perception failure case study in Section D.3. Fig (left to right): example of Failure-1 (distance), Failure-2 (poor light) and Failure-1 and Failure-2 both in one scene (distance and poor light), respectively. Bounding boxes for detected objects (misdetections) shown in yellow (red) with detection confidence numbers. Each scene has two cars and a pedestrian.

- Goal: Identify the subset of active sources from a set of six potential sources with known locations.
- Ground Truth: Sources 3-8 were used, with Sources 3, 5, 6, and 8 were set as active.
- Maximum Sensor Threshold ($y_{max}$): 30.0 hits/second.
- Hypothesis Space: The set of all possible on/off combinations for the six sources. This is a discrete space with $2^6 = 64$ unique hypotheses.
- Degree of curiosity: $\beta = 5.0$.

## D.3  CONSTRAINED ACTIVE SEARCH

### D.3.1  PERCEPTION IN SELF DRIVING SIMULATION CARLA

We consider the failure discovery for YOLO-based object detection (Jiang et al., 2022; Redmon & Farhadi, 2018) in the CARLA simulator (Dosovitskiy et al., 2017).

This requires the generation of various scenarios in the environment using CARLA simulator. The environment is a composition of a static context and scenario $\phi$. We use the probabilistic programming framework Scenic (Fremont et al., 2019) for sampling scenarios with varying contextual information for a fixed scenario variable $\phi$. For the simulations presented in this paper, we used a publically available, pre-existing environment (Dreossi et al., 2019), which consists of the ego vehicle maneuvering on the road with two non-ego agents– two non-ego cars and a pedestrian crossing the road. We use YOLO object detection model to detect all non-ego agents in a scene. The generated scenario is seeded for reproducibility, so that for a given scenario $\phi$, the environment can be treated as a deterministic quantity. Each scenario is defined using $\phi = [b_e, b_l, s]$, where $b_e, b_l \in [5, 15]$ represent the braking threshold of the ego car and lead car (non-ego car in-front of ego car) (m) and $s \in [0, \pi/2]$ denotes the sun altitude angle (rad). Each of these quantities is normalized to be within $[0, 1]$, and the normalized scenario is chosen as the decision variable $x$ for active inference.

We are interested in environment variables (scenarios) that lead to two specific types of failures– failure to detect non-ego agents due to large distance from ego vehicle (Failure 1), and failure to detect non-ego agents due to poor scene lighting (Failure 2). Fig. 5 shows examples of the discussed object detection failures we aim to discover.

### D.3.2  LLM-BASED EVALUATION FOR CARLA

We use LLM to perform evaluations for whether a generated scene corresponds to a failure due to specific type.

Each evaluation for a specific scenario corresponds to $T = 60$ steps of simulations. Images recorded from the camera view are used for object detection and classification at every 10 steps using YOLO-v3 (Redmon & Farhadi, 2018), and the classified image are used as inputs to GiT (Wang et al., 2022) to predict a likely failure type based on fine-tuning data.

Results obtained from YOLO along-with the reports from GiT are used as inputs to GPT3 model for failure evaluation, which is queried 6 times per evaluation, and assigns binary scores pertaining to each failure mode for each scene (camera image). The average value reported across 6 scenes is used to construct $c_i : \mathcal{Z} \to \mathbb{R}$ for $i = 1, 2$ as: $c_i(z) = \frac{1}{T} \sum_{t=1}^{T} b_t^i$. Here $b_t^i \in \{0, 1\}$ is a scene-specific binary evaluation provided by the LLM based on report generated by GiT to assess if an object detection failure is observed in a given scene and corresponds to Failure-$i$.

We use GPT-3.5 Turbo model for LLM-based binary evaluations, with each evaluation we query the LLM 6 times and combine the binary evaluations for all 6 runs. Note that usage of LLM is not a core part of our methodology and is only used to as a subjective evaluator.

We show the prompt used for failure evaluation of each scene using the LLM in the box below. Information shown in red and blue is obtained from GiT captioning system, and CARLA respectively. The output of the LLM is used to obtain a binary number for each scene which is composed to give a scenario specific cost function $c_1, c_2$.

---

**Prompt Used for CARLA evaluation**

You will be provides the analysis of YOLO Object detection on an image that was taken from the camera feed of CARLA simulator. The simulator is simulating a pedestrian crossing the road before a car in front of the ego car. There are two cars and one pedestrian in each image. The information provided:

1. *Objects detected*: List of objects detected by YOLO in the image. This list should have atleast one object from the *Objects to detect* list

2. *Objects to detect* list:

    - One object with one of the following labels: 'car','truck',
    - One object with one of the following labels: 'car', 'truck','bus','motorcycle','bicycle'
    - One object with one of the following labels: 'person'

3. Reason: The reason is a brief explanation of the failure to detect all objects, if that happens, and is generated by a pre-trained GiT model in the form of captions for the image.

We are looking to discover images where YOLO fails to detect an object due to **bad light** and/or l**arge distance.** If the list of *objects detected* has an object missing from the *objects to detect* list, look at the reason. The reason can have other components as well, but it can 'only' be considered as **bad light** if at least one of the objects was failed to detect strictly due to **bad light**. Similarly, the reason can have other components as well, but it can 'only' be considered as **large distance** if the reason contains the phrase 'far away'. Follow the response instructions while responding.
Response Instructions: Respond should be an integer 0, 1, 2, 3 or 4:

    - 0 indicating that at least one object was missing from the 'objects to detect' list, but the reason provided does not correspond to bad light or large distance.

    - 1 indicating that an object was not detected and the reason provided corresponds to bad light only.

    - 2 indicating that an object was not detected, and the reason corresponds to large distance only.

    - 3 indicating that an object was not detected, and the reason corresponds to both large distance and bad light.

    - 4 indicating all objects are detected. Do not provide explanation.

Response format: Response: [integer], where integer = 0,1,2,3,4.
The list of objects detected and reason for incomplete detection for the image are as follows:

    - Objects detected: {objects}

    - Reason: {reason}

### D.3.3 TASK-SPECIFIC CONFIGURATIONS

- Goal: We consider two cost functions $c_1, c_2$ associated with each type of failure. The goal is to sample from the set $\Omega = \{z|c_1(z) \geq C_1, c_2(z) \geq C_2\}$, and we consider three target sets defined by $C_1 = C_2 = 0.1$, $C_1 = C_2 = 0.5$, and $C_1 = C_2 = 0.8$.
- Degree of curiosity: $\beta = 20.0$.

## D.4 COMPOSITE BAYESIAN OPTIMIZATION

### D.4.1 PREFERENCE EVALUATION

We simulate human-in-the-loop or policy-driven decision-making via pairwise preference queries. That is, for selected pairs of outcomes $(y_1, y_2)$, a preference function indicates which design is preferred. These preferences are generated based on a latent utility function, not revealed to the optimizer.

An initial set of 1 pairwise preferences is randomly sampled to initialize the model. Each step of the optimization selects new pairs to query, guided by the used acquisition strategy.

### D.4.2 TASK-SPECIFIC CONFIGURATIONS

**Vehicle Safety.**

- Goal: Optimize vehicle crash-worthiness.
- Testbed: See (Tanabe & Ishibuchi, 2020) for details.
- Ground Truth Preference: $g(y) = -(y - y^*)^2$, where $y^* = [1864.7202, 11.8199, 0.2904]$.
- Degree of curiosity: $\beta = \gamma = 1.0$.

**Penicillin.**

- Goal: Maximize the penicillin yield while minimizing time to ferment and the CO2 byproduct.
- Testbed: See (Liang & Lai, 2021) for details.
- Ground Truth Preference: $g(y) = -(y - y^*)^2$, where $y^* = [25.935, 57.612, 935.5]$.
- Degree of curiosity: $\beta = \gamma = 1.0$.

| Performance Metrics | Definition |
|---|---|
| Voltage Fairness | Measures the variance in bus voltages across the network; lower variance implies more equitable voltage delivery. |
| Total Cost | Combines capital expenditures for DER installation and operational costs related to reactive power support. |
| Priority Area Coverage | Quantifies the share of power delivered to high-priority buses, such as rural or underserved regions. |
| Resilience | Assesses the percentage of time that all bus voltages remain within safe operating limits under perturbations (*e.g.*, load uncertainty or line outages). |

**Energy Resource Allocation.**

- Goal: Identify deployment strategies for Distributed Energy Resources (DERs) in Optimal Power Flow (OPF) that align with implicit ethical preferences across multiple performance dimensions detailed in the following table.
- Testbed: IEEE 30-bus network in pandapower library.
- Ground Truth Preference: $g(y) = a^\intercal y$, where $a = [1, -1, 2, 1]$
- Degree of curiosity: $\beta = \gamma = 1.0$.

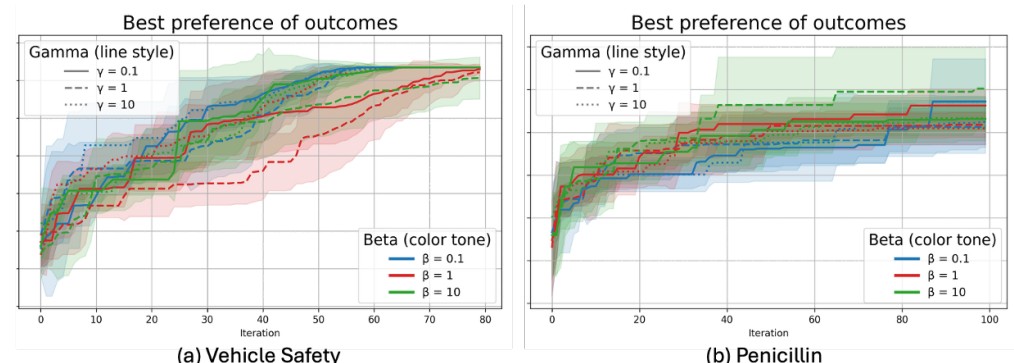

(a) Vehicle Safety  (b) Penicillin

Figure 6: Ablation studies on $\beta$ and $\gamma$ for vehicle safety and penicillin. Error bars represent $\pm 1$ std over 5 seeds.

### D.4.3 ABLATION STUDIES ON $\beta$ AND $\gamma$

We further conduct an ablation study to investigate how the curiosity weights $\beta$ and $\gamma$ affect the learning dynamics. Recall that $\beta$ controls the mutual information term for the preference model $g(y)$, while $\gamma$ controls the mutual information term for the outcome model $f(x)$. Figure 6 reports the evolution of the best observed preference value $g(y)$ over iterations, with line style encoding $\gamma$ and color encoding $\beta$.

For the penicillin task (right), performance is clearly sensitive to both $\beta$ and $\gamma$. Runs with a small preference-curiosity weight ($\beta = 0.1$, blue) consistently underperform those with larger $\beta$, indicating that actively reducing uncertainty in the preference model is crucial. Increasing $\beta$ to 10 (green) yields the highest final preference values, especially when combined with moderate or large outcome curiosity ($\gamma = 1$ or $\gamma = 10$); these settings converge faster and reach higher plateaus than configurations with $\gamma = 0.1$. This suggests that, in problems where the mapping from outcomes to preferences is non-trivial, it is beneficial to invest in both exploring the outcome space and actively refining the preference model.

For the vehicle safety task (left), the effect of $\beta$ and $\gamma$ is more modest. Most configurations eventually reach similar best preference values, and the differences appear mainly in the early iterations: larger $\gamma$ (dotted lines) tends to accelerate initial improvement, whereas small $\beta$ (blue) already suffices to achieve desired exploration. Overall, this task is more robust to the precise choice of $\beta$ and $\gamma$, consistent with an easier or more informative outcome–preference structure.

### D.4.4 EXTENDED BASELINE COMPARISON WITH BOPE

To highlight the benefits of jointly learning and optimizing, rather than separating these into stages, we extend the baseline comparison with **BOPE** from Lin et al. (2022).

The original BOPE framework is intentionally flexible and leaves many problem-specific design choices open, especially regarding how and when to switch between preference exploration and experimentation. In our comparison, we consider four representative stage-wise variants:

- **BOPE-I**: A two-phase strategy that starts with qEUBO (preference-focused exploration) and switches to qNEI in the second half (objective-driven refinement), illustrating the effect of premature exploitation.

- **BOPE-II**: A two-phase strategy that starts with qNEI (exploring the objective space) and switches to qEUBO in the second half (exploiting the learned preference model), using a frozen outcome snapshot for qEUBO.

- **BOPE-III**: A qEUBO-only variant where experiments are selected by qEUBO with a newly sampled objective realization at each iteration, encouraging stronger exploration through objective variation.

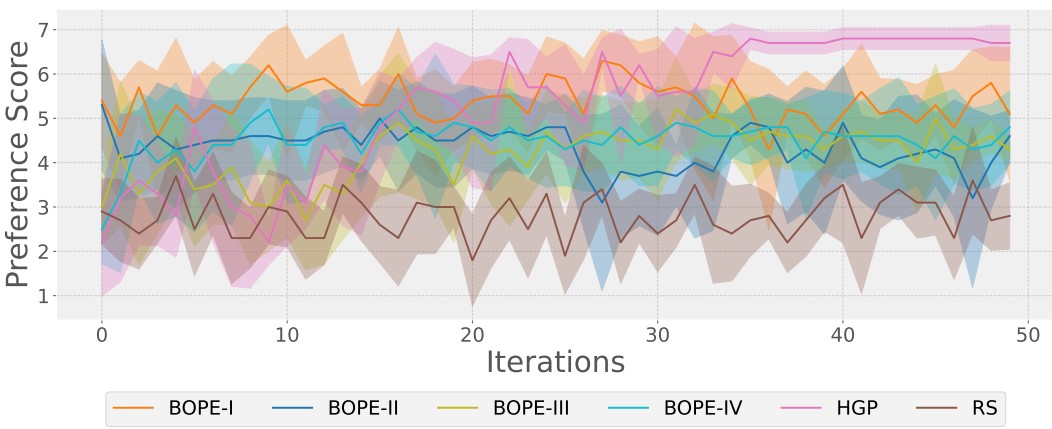

Figure 7: Extended baseline comparison with BOPE for energy resource allocation. Error bars represent ±1 standard deviation over 5 seeds.

- **BOPE-IV**: A BOPE variant that uses standard preference exploration (qEUBO) and selects experiments exclusively with qNEI, fully refitting both outcome and preference GPs after each update.

Figure 7 reports their preference scores over random sampling (**RS**). It is evident that our active-inference method for hierarchical GPs (**HGP**) consistently discovers higher-preference regions after a brief initial exploration phase, while BOPE variants are highly sensitive to their stage-wise design choices. **BOPE-I**, being preference-driven in the first phase, initially attains higher preference values but fails to balance exploration and exploitation, leading to poor convergence in the second half; its performance is also sensitive to the precise switching point. **BOPE-II** explores first and then optimizes, achieving better final performance than BOPE-I, but the strict separation between exploration and exploitation still yields suboptimal outcomes. **BOPE-III** mixes both aspects but remains more exploitation-centric, performing better than BOPE-I/II yet still below HGP. **BOPE-IV** and HGP share the idea of refitting both models at each step, but BOPE-IV converges to a lower-preference solution. In contrast, our acquisition strategy jointly leverages information from both the outcome and preference models at every iteration, leading to higher sample efficiency and more reliable discovery of high-preference regions.

# E  USE OF LARGE LANGUAGE MODELS

We use LLM to revise and polish drafts of the introduction, abstract, conclusion, and limitations sections to enhance clarity, flow, and accessibility for a broader audience.

We also use LLM as an evaluator for the experiments of failure discovery in CARLA, which is detailed in Appendix D.3.

