# OpenReview forum: "Pragmatic Curiosity: Unifying Bayesian Optimization and Experimental Design via Active Inference"
_ICLR.cc/2026/Conference — Submitted to ICLR 2026_

### Official Review · Reviewer_xcCu · 2025-10-22

**Soundness:** 3
**Presentation:** 3
**Contribution:** 3
**Rating:** 6
**Confidence:** 4

**Summary:**

This paper bridges Bayesian Experimental Design (BED), which focuses on learning unknown environmental parameters and Bayesian Optimization (BO), which focuses on finding optimal configurations, with the theory of Active Inference (AIF). It interprets the expected free energy as the sum of information gain (learning) and surprisal (decision-making).

Based on this insight, the authors propose a new acquisition strategy called _pragmatic curiosity_, which balances exploration and exploitation in a principled manner without manual tuning. Experiments demonstrate its effectiveness across tasks involving explicit and black-box learning and decision-making goals.

**Strengths:**

- The authors provided theoretical intuitions about the connection between BED and BO. They leveraged AIF, a framework common in computational neuroscience, to formulate their results. They also validated their findings in the UCB acquisition of BO. This provides new insights to the BED, BO, and AIF communities.

- The authors further designed a novel decision-making strategy based on the theoretical findings and validated it through experiments. This provides solid evidence for the proposed framework.

**Weaknesses:**

- The formulation of AIF is not clear, especially the relationship between the "true model" $p(\cdot)$ stated in Line 161 and the prior preference distribution over the outcome $y$, $p(y)$.

    - Usually, we assume a prior distribution of $f(x) \sim p(y|x)$. In Line 648, $p(y|x)$ is simplified as $p(y)$. Is it because it is assumed that $p(x) \propto 1 \implies p(y|x) = p(y)$?

- I have a concern in the technical details. While the authors assume a surrogate $q(s)$ for parameters $s$ whose real distribution is $p(s)$ in Equation (1), this assumption is **intentionally broken** in the derivation of the form of expected free energy in Line 668 by assuming $q(s) = p(s)$. Thus, the form of expected free energy is just an approximation. How could the authors justify this?

- There is an issue in the experimental section. The authors claim to have compared their method with standard BO in Sec. 5.1 and Sec. 5.2, but the BO results do not appear in the corresponding figures.

- The framework introduces temperature/curiosity parameters ($\beta, \gamma$, etc.). The paper claims that the balance **emerges** from beliefs, yet experiments still set $\beta, \gamma$ per task (Appendix lists the values). It remains unclear how sensitive performance is to these settings and whether the method truly removes manual tuning in practice. The authors should provide sensitivity plots or guidance.

**Questions:**

see weakness

---

> ### Author Response · Authors · 2025-11-23
> **Response to Reviewer xcCu (Part 1/2)**
>
> We appreciate the reviewer’s thoughtful comments and for recognizing that our work offers new insights to the BED, BO, and AIF communities, supported by solid experimental evidence. We hope that the revisions to the manuscript and the responses below address your concerns.
>
> ## Summary
>
> In brief, we have:
>
> 1. **Clarified the technical details** of the AIF formulation and derivations, particularly the roles of $p(y|x)$ and the prior preference distribution $p(y)$, and the use of surrogate models $q(s)$.
> 2. Clarified that we adopt and report a **BO-type (greedy) baseline** tailored to each task, and how this relates to "standard BO."
> 3. Added **two ablation studies** on the curiosity parameters $\beta$ and $\gamma$, and provided practical guidelines on how to choose them in different settimgs.
>
> ## Detailed Reply
>
> >  Weakness 1: AIF formulation and the roles of $p(y|x)$ and $p(y)$
>
> We agree that the distinction between the true model and the preference distribution needs to be explicit.
>
> In the revised text, we clearly state that:
> * $p(y|x)$ is the true outcome distribution conditioned on the decision $x$, and
> * $p(y)$ is a prior preference over outcomes that does not depend on $x$.
>
> The AIF objective is designed so that, when we minimize it, we favor decisions $x$ that make the predictive outcome distribution align with the preferred outcomes. Concretely, in the derivation in Appendix A (lines 766-767) we now directly write the preference term as $- \mathbb{E}_{q(y|x)} \log p(y)$. This makes it clear that we are *evaluating* the preference $p(y)$ under the predictive model $q(y|x)$, rather than assuming $p(y|x)=p(y)$.
>
> Intuitively, the agent asks: "If I take action $x$, what outcomes will I likely see, and how much do I like those outcomes according to $p(y)$?"
>
> >  Weakness 2: Approximate nature of the EFE and the use of $q(s)$
>
> Good observation! You are definitely right that the EFE we used in equation 4 is just an approximation, but it is the best tractable approximation we can have based on the currently available information.
>
> The full expression includes a divergence term between the surrogate model and the true model, which depends on the intractable posterior $p(s|x,y)$. In practice, the agent's best available model at time $t$ is the posterior given the current data  $q(\cdot)=p(\cdot|\mathcal{D}_{t})$.
>
> In the revised Appendix A (lines 789-791), we now say this explicitly: we treat $q(\cdot)=p(\cdot|\mathcal{D}_{t})$ as our best approximation of the true model $p(s)$, i.e., $q(s) \approx p(s)$.Under this approximation, the divergence term effectively disappears and we obtain the tractable EFE expression in Eq. (4).
>
> Conceptually, this is also aligned with the decision-theoretic view: the agent cannot act based on an unknown ground-truth distribution, only on its current belief. Our EFE therefore defines decisions that are Bayes-optimal with respect to the surrogate belief $q(\cdot)=p(\cdot|\mathcal{D}_{t})$. As more data are collected, this belief converges towards the true model under standard assumptions, so the approximation error shrinks over time.
>
> >  Weakness 3: BO baselines in Sections 5.1 and 5.2
>
> We understand the confusion and appreciate the chance to clarify.
>
> In Sections 5.1 and 5.2, we *do* compare against BO-type baselines. These are implemented as greedy strategies tailored to each task and are labeled as “Greedy” in the figures.
>
> To avoid ambiguity, we now explain at the beginning of Sec. 5 (lines 328-330) that the problems we consider are not canonical single-objective BO or pure BED tasks. Each task has both a learning goal (e.g., identifying parameters or improving coverage in the parameter space) and an optimization goal (e.g., respecting safety constraints or improving coverage in a metric space).
>
> For each task, we define: (i) a BO-type baseline focusing on the optimization aspect (the Greedy strategy), and (ii) a BED-type baseline focusing on learning, typically via expected information gain.
>
> For example:
>
> * In **constrained system identification** (Sec. 5.1, lines 353-355), the BO-type baseline chooses points that minimize the probability of constraint violation, while the BED-type baseline maximizes information gain about the unknown parameters.
> * In **targeted active search** (Sec. 5.2, lines 451-453), the BO-type baseline maximizes coverage in the metric space, and the BED-type baseline maximizes coverage in the parameter space.
>
> We now describe these baselines explicitly in the text so that the connection to BO-type behavior is transparent, even though we do not directly plug in EI/UCB/MES, which assume canonical BO objectives that do not match these intertwined tasks.

---

> > ### Author Response · Authors · 2025-11-23
> > **Response to Reviewer xcCu (Part 2/2)**
> >
> > >  Weakness 4: Sensitivity to curiosity parameters $\beta$ and $\gamma$
> >
> > We fully agree that understanding the role of the curiosity parameters is important for practical use.
> >
> > In the revised version, we have:
> > 1. Added ablation studies where we systematically vary $\beta$ (and $\gamma$ in Section 5.3).
> > 2. Summarized practical guidelines that emerge from these experiments.
> >
> > The main takeaways are:
> >
> > * In **Sec. 5.1 (Choices of $\beta$, lines 364-373)**, we observe that the "best" $\beta$ depends on how informative the measurements are about the latent parameters. When measurements are less informative, the mutual information term becomes smaller, and increasing $\beta$ compensates by amplifying the information-gain component. If $\beta$ is too large, the policy becomes overly exploratory and performance degrades, which matches our intuition. In practice, $\beta$ should be just large enough for the curiosity term to matter, but not so large that it induces unnecessary exploration. (See Figure 2 in the main text)
> > * In **Sec. 5.3 (Choices of $\beta$ and $\gamma$, lines 513-520)**, we study the joint effect of $\beta$ and $\gamma$. When the preference model $g(y)$ is complex or poorly informed by initial data, a larger $\beta$ helps the method allocate more queries to reduce uncertainty in $g$, while a moderate-to-large $\gamma$ preserves sufficient exploration of the outcome model $f(x)$. In simpler or well-informed settings, smaller values of $\beta$ and $\gamma$ already work well, and performance is relatively robust to their exact values. (See Figure 6 in Appendix C.4.3)
> >
> > Overall, these results show that the method is not overly fragile with respect to $\beta$ and $\gamma$, and we provide concrete, task-driven guidance for choosing them in practice.

---

> ### Comment · Reviewer_xcCu · 2025-11-26
> **response**
>
> I appreciate the authors' effort to clarify my misunderstanding and to make the presentation better. I maintain my positive score.

---

### Official Review · Reviewer_BqG3 · 2025-10-31

**Soundness:** 4
**Presentation:** 3
**Contribution:** 4
**Rating:** 8
**Confidence:** 2

**Summary:**

The paper proposes a framework called pragmatic confusion, combining the goal-oriented objectives of Bayesian Optimization (BO, where the goal is the maximize an objective), and Bayesian Experimental Design (BED, where the goal is to gain maximal information about unknown system parameters). The framework is based on active inference, which attempts to minimize the expected free energy, leading to a balance between pragmatic (goal-seeking) and epistemic (information seeking) objectives, and includes various standard BO and BED frameworks (eg GP-UCB and EI) as special cases.

**Strengths:**

The paper is quite well written. I found the underlying idea compelling, and as best as I can tell the mathematics is accurate. The idea matches quite well with the motivation for GP-UCB, but generalized and systematised to cover a much wider range of problems than is typical for BO alone. Experimental results are impressive for the range of problems covered.

**Weaknesses:**

In general, I do wonder about the computational cost required to maximize the acquisition function, but experiments appear to indicate that the approach is practical.

**Questions:**

None.

---

> ### Author Response · Authors · 2025-11-23
> **Response to Reviewer BqG3**
>
> We appreciate the reviewer for acknowledging both the merits of our underlying idea and the strength of the experimental results!
>
> Regarding computational cost, you are absolutely right that estimating the EFE can be a central bottleneck in practice. For problems of moderate complexity, such as those considered in this paper, the standard Monte Carlo samplers built into BoTorch provide satisfactory performance, with computational cost comparable to other mutual-information-based acquisition functions (e.g., EIG, ES).
>
> That said, for high-dimensional or time-critical applications, more scalable and efficient approximations of the EFE will be needed to handle complex real-world problems. We now explicitly highlight this as an important direction for future work in Section 6.

---

### Official Review · Reviewer_ZUrj · 2025-10-31

**Soundness:** 2
**Presentation:** 2
**Contribution:** 2
**Rating:** 2
**Confidence:** 3

**Summary:**

The authors offer a unifying framework for two historically different fields, Bayesian Optimization and Bayesian Experimental Design. Although the common wisdom takes these for being almost identical, being able to precisely characterize in what sense this is the case is a more difficult task. To that end, this submission introduces ``pragmatic curiosity'', a new paradigm that offers to resolve the exploration-exploitation dilemma across BO and BED, by the minimization of a sole objective called Expected Free Energy.  After introducing their theoretical framework and deriving an acquisition criterion from it, the authors apply their approach to a range of experiments from constrained system identification, targeted active search, and composite optimization with unknown preferences.

**Strengths:**

- The paper recasts the exploration/exploitation dilemma in active learning/Bayesian experimental design/Bayesian optimization as a tradeoff between information-seeking behavior, and goal-directed behavior.
- On the experiments selected by the authors, the proposed approach yield satisfactory results.

**Weaknesses:**

This might be entirely on me, but I found the paper to be badly written. While the theoretical framework and the general intuition were understandable to me, the execution, meaning the practical computations and the experiments on test problems, is a major flaw to me.
I believe there are simply too many things lacking, and too many things in the Appendix that should have been in the main text.

- For each problem, I think we should be able to tell __from the main text__ its dimensionality: how many variables are we optimizing for? This should not be part of the appendix. It could be as simple as having "Source Localization 2D" in the xlabel of Figure 1 for instance. For Fig1/Sec 5.1 & Fig2/Sec 5.2, input dimensionality & hypothesis space could be retrieved from the appendix, but for Fig3/Sec 5.3, I could not find it. I assume it is well-described in the reference mentioned therein, but having to look through the text of a reference mentioned in the appendix of your paper to find something as simple as problem input dimensionality seems cumbersome.

- While I could see (e.g., through Table 1) the connection with BO, it was more difficult to pinpoint how this work relates to BO during the experiments section. For instance, the last example, section 5.3, involves composite BO. It is mentioned that "since the setting in this category does not follow the traditional BO or BED setting, there is no baseline that can be directly used". What about the paper which you mentioned then, [1]? Can you explain why this approach could not be employed here? From a more general point of view, for a paper that states in its title "unifying Bayesian optimization and experimental design", I would have expected to also see classical BO acquisition functions, like EI, UCB, or MES, being compared.

[1] Preference Exploration for Efficient Bayesian Optimization with Multiple Outcomes, AISTATS 2022.

- The practical computation/estimation of equation 7 is lacking.

- I believe a substantial amount of literature could have also been mentioned here:

[4] Maximizing acquisition functions for Bayesian optimization, NeurIPS 2018, shows how many classical BO acquisition functions can be written as the expectation of specific utility functions $u$ under a posterior distribution $p(y|x, \mathcal{D})$, I think this is worth mentioning, specifically how your Table 1 compares to their Table 1.

[5] Prediction-Oriented Bayesian Active Learning, AISTATS 2023, introduces a new state-of-the-art Bayesian Experimental Design criterion that allows going beyond purely reducing parameter uncertainty, which can be suboptimal from a predictive performance standpoint. This criterion has been since then adopted in many other studies since then, and in my opinion, it should be added to the baselines.

[6] Self-Correcting Bayesian Optimization through Bayesian Active Learning, NeurIPS 2023, shows how active learning and BO can be related by leveraging classical active learning criteria to reduce the uncertainty in GP surrogate hyperparameters in BO. They propose an acquisition function that balances this reduction in hyperparameter uncertainty while also seeking promising candidates; this is the very tradeoff illustrated in this submission. At the very least, this should be mentioned.

I did not run it myself, but I assume a deep search run with any language model would have surfaced these publications, and perhaps more. Actually, it just occurred to me that the paper does not contain a related work section. The introduction is quite dense and contains related work, but only to a limited extent.

I might be wrong on several things here, and upon genuine misunderstanding from my side and clarifications from the authors on the weaknesses (and questions below), I would consider raising my score.

**Questions:**

- Can you provide a rationale for increasing the degree of curiosity $\beta$ in the experiments depicted in Sec 5.1 $\beta$ goes from 0.5 to 1 to 5 depending on the setting. More generally, carrying an ablation study where $\beta$ is varied would have been helpful.

- In preliminaries, Section 2.1, the acquisition function is defined as taking values in $\mathbb{R}_+$, this need not be the case, for instance with GP-UCB if the GP surrogate takes negative values?

- How is the saturation threshold $C(y)$ defined in Section 5.1 computed?

---

> ### Author Response · Authors · 2025-11-23
> **Response to Reviewer ZUrj (Part 1/3)**
>
> We thank the reviewer for carefully reading our paper and for acknowledging the importance of the problem and the quality of our experimental results. We have revised the paper to improve clarity, expanded the experimental section, and added new ablation studies. We believe these changes substantially strengthen the manuscript and address **all** your concerns. Please let us know if you have further questions.
>
> ## Summary of Changes
>
> In brief, we have
>
> 1. **Clarified and reorganized the experimental section.** We moved key experimental details from the appendix into the main text and added a description of the practical computation of the acquisition function using BoTorch.
> 2. **Added a dedicated related-work subsection (Sec. 2.3)** discussing prior work on the synergy between BO and active learning, including the references you suggested.
> 3. Included **new experiments** in Sec. 5.3 that **compare our method against the baselines proposed in [1]** (“Preference Exploration for Efficient Bayesian Optimization with Multiple Outcomes,” AISTATS 2022).
> 4. Clarified why classical BO acquisition functions such as EI, UCB, or MES are not directly applicable to our problem settings in Sec. 5.1 and Sec. 5.2, and **explained more clearly** how we construct BO-type (optimization-focused) and BED-type (learning-focused) baselines tailored to each task.
> 5. Added **two new ablation studies** analyzing how the curiosity parameters $\beta$ and $\gamma$ affects performance, and provided practical design guidelines based on these results.

---

> > ### Author Response · Authors · 2025-11-23
> > **Response to Reviewer ZUrj (Part 2/3)**
> >
> > ## Detailed Reply
> >
> > > Weakness 1: Problem dimensionality and clarity of experiments
> >
> > In line with your comment that the paper’s exposition could be improved, we have edited the experiments section for readability and structure, with more explicit descriptions of the chosen baselines (lines 328-330), input and output dimensionality for each task (line 348, 430, 481), and details about practical computation (line 331).
> >
> > > Weakness 2: Relation to BO and comparison to classical BO acquisitions
> >
> > **(a) Comparison with [1] in Sec. 5.3**
> >
> > In the revised manuscript, Sec. 5.3 (Benefits of joint learning and optimization, lines 521-529) now includes a dedicated comparison with the BOPE framework of [1] under the same preference-learning and composite-outcome setting (see also Appendix C.4.4 for full experimental details and plots).
> >
> > Conceptually, BOPE separates preference exploration and experimentation into stages and relies on design choices (e.g., when to switch stages and how often to update models) to decide when to explore versus optimize. In contrast, our active-inference acquisition unifies exploration and exploitation of both the outcome and preference models at every step. In fact, in its best-performing configuration (switching stages and updating models at each iteration), the BOPE acquisition reduces to a special case of our nested objective in (10): it corresponds to setting $\beta=\gamma=0$, thereby omitting the information-gain terms.
> >
> > Empirically (Figure 7 in Appendix C.4.4), our full method consistently discovers higher-preference regions than these BOPE variants and is markedly less sensitive to stage-wise design choices. The ablation baselines we include in Figure 4 (removing individual terms from our acquisition) further confirm that jointly accounting for both outcome and preference information gain leads to more reliable performance, while BOPE-style staged approaches require careful manual tuning to achieve good results.
> >
> > **(b) Relation to BO in Sec. 5.1 and Sec. 5.2 and choice of baselines**
> >
> > You are correct that the problems we study originate from the BO/BED literature, and we now make this connection more explicit at the beginning of Sec. 5 (lines 328-330). In particular, we clarify that:
> >
> > * **Constrained System Identification (Sec. 5.1)** couples: (i) a learning objective (accurately inferring unknown system parameters), and (ii) an optimization objective (respecting safety constraints during data collection).
> > * **Targeted Active Search (Sec. 5.2)** couples: (i) a learning objective (improving coverage in the parameter space), and (ii) an optimization objective (improving coverage in the outcome/metric space).
> >
> > These tasks do not fit the canonical setting of BO (optimizing a single scalar objective over $\mathcal{X}$) nor of standard BED (pure information gain about parameters). Classical BO acquisition functions such as EI, UCB, or MES are designed for that canonical BO objective and do not directly capture the intertwined learning–optimization objectives and constraints in our tasks.
> >
> > Instead, for each problem we construct:
> >
> > * **BO-type baselines** that are optimization-focused (e.g., targeting performance or safety-related quantities), and
> > * **BED-type baselines** that are learning-focused (e.g., maximizing information gain about parameters).
> >
> > We have revised the beginning of Sec. 5 (lines 328-330) to explicitly state this design choice and to detail what “BO-type” and “BED-type” mean for each specific task (lines 353-355, 451-453). This is intended to make the connection to BO clearer while ensuring that all baselines are well-defined and fair for the non-canonical tasks we consider.
> >
> > >  Weakness 3: Practical computation/estimation of Equation (7)
> >
> > We appreciate this comment and agree that the practical side of Eq. (7) should be clearer. We first emphasize in the text that Eq. (7) provides an abstract, task-agnostic expression of the acquisition. For each task, we then instantiate Eq. (7) as concrete forms (Eqs. (8), (9), and (10) in the paper). In the revised version, we explicitly explain that these instantiated forms are evaluated via Monte Carlo estimation using the built-in samplers in BoTorch (line 331).
> >
> > We added a short description of this implementation in the main text (line 331) and provided further details in Appendix C.1 (Simulation environment, lines 916-917), including that the Monte Carlo samples are drawn from the posteriors for each model to approximate the expectations in Eq. (7).

---

> > > ### Author Response · Authors · 2025-11-23
> > > **Response to Reviewer ZUrj (Part 3/3)**
> > >
> > > >  Weakness 4: Missing related work section and comparison with [5]
> > >
> > > **(a) Related work subsection**
> > >
> > > We have added a dedicated related-work subsection (Sec. 2.3, The synergy between BO and BED, lines 144-161). This subsection summarizes prior work on BO, BED, and their intersections, and highlights existing approaches that attempt to combine optimization and learning objectives, including the references you suggested.
> > >
> > > This change separates background and related work from the introduction, making both sections clearer and less dense.
> > >
> > > **(b) Comparison with [5]**
> > >
> > > Regarding [5], we now clarify why we do not include it as a separate baseline in our experimental section. The method in [5] is designed for scenarios where there are specific inputs of interest, and the goal is to directly reduce predictive uncertainty at those designated points. In our problem settings, however, there is no preference over particular input locations: all inputs are treated symmetrically.
> > >
> > > In such a symmetric setting, [5] effectively reduces to a standard expected information gain (EIG) criterion over the parameter space, which is already included as one of our baselines.
> > >
> > > >  Question 1: Rationale for increasing $\beta$ and ablation study
> > >
> > > Thank you for this suggestion. We have made two changes in response:
> > >
> > > **(a) New ablation studies and qualitative insights:**
> > >
> > > We added ablation experiments where we systematically vary $\beta$ (and $\gamma$ in Sec. 5.3) to study how the curiosity terms influence performance and exploration–exploitation trade-offs. The results, reported in Sec. 5.1 (Choices of $\beta$, lines 364-373) and Sec. 5.3 (Choices of $\beta$ and $\gamma$, lines 513-520), show that moderate curiosity generally improves performance, while overly large values lead to excessive exploration and degraded accuracy.
> > >
> > > **(b) Rationale and Design guidelines:**
> > >
> > > In the main text, we now explain the rationale behind the choice and scaling of $\beta$ and $\gamma$.
> > >
> > > In Sec. 5.1 (Choices of $\beta$, lines 364-373), we highlight that as the correlation between direct sensor measurements and the latent parameters weakens across tasks (a)–\(c\), the magnitude of the mutual information term in Eq. (8) decreases. Increasing $\beta$ compensates for this reduced informativeness by rescaling the information-gain term. However, if $\beta$ is too large, the acquisition becomes overly exploratory and harms estimation performance. In practice, $\beta$ should be just large enough for the curiosity term to matter, but not so large that it induces unnecessary exploration. (See Figure 2 in the main text)
> > >
> > > In Sec. 5.3 (Choices of $\beta$ and $\gamma$, lines 513-520), we summarize the ablation findings as practical guidelines: when the preference model $g(y)$ is complex or poorly informed by the initial data, a larger $\beta$ helps by explicitly allocating queries to reduce uncertainty in $g$, while $\gamma$ should be kept in a moderate-to-large range to maintain sufficient exploration of the outcome model $f(x)$. In simpler or more informative tasks, smaller values of $\beta$ and $\gamma$ already suffice, and the method becomes less sensitive to their precise tuning. (See Figure 6 in Appendix C.4.3)
> > >
> > > >  Question 2: Codomain of the acquisition function
> > >
> > > You are absolutely right. In the revised version, we corrected the definition to $\alpha:\mathcal{X} \mapsto \Re$ in Sec. 2.1 (line 121). This was a notational typo; the subsequent derivations and analysis do not rely on $\alpha$ being nonnegative, so the correction does not affect any results.
> > >
> > > >  Question 3: Definition of $C(y)$ in Sec. 5.1
> > >
> > > Thank you for catching this missing detail. In the revised Sec. 5.1 (line 349), we now explicitly define how the saturation threshold is implemented in our 2D plume-field environmental monitoring experiments: "We perform experiments on a real-world environmental monitoring problem in 2d plume fields where the sensors have a saturation threshold $y_{max}$ (i.e., $C(y)=y-y_{max}$)."

---

> > > > ### Comment · Reviewer_ZUrj · 2025-11-27
> > > >
> > > > Thank you for the detailed rebuttal. I really appreciate the substantial amount of work you put into revising the paper.
> > > > The additional experiments and clarifications have addressed several of my concerns. As a result, I will increase my score from 2 to 4.
> > > >
> > > > However, I still feel that the manuscript would benefit from more rounds of polishing for presentation/how the main message is conveyed. Unifying BO and BED is a non-trivial goal, and while the revision moves in the right direction, I am not yet convinced that the current version fully delivers on this promise. I do not think the paper is yet at the level of a weak accept (6), __but if there were a “5” rating option, I would choose it__. I realize my comment is far from actionable. It mostly reflects the (subjective) hesitation that keeps me from going higher than a 4.

---

> > > > > ### Author Response · Authors · 2025-12-03
> > > > > **2nd Response to Reviewer ZUrj**
> > > > >
> > > > > $$
> > > > > \def\a{\color{#648FFF}{\textsf{ZUrj}}} \def\b{\color{#E69F00}{\textsf{BqG3}}} \def\c{\color{#DC267F}{\textsf{xcCu}}}
> > > > > $$
> > > > >
> > > > > Thank you again for your thoughtful follow-up and for raising your score. We truly appreciate the time and care you have put into reviewing our work.
> > > > >
> > > > > You are absolutely right that **unifying BO and BED is a non-trivial and long-term goal**. We fully agree that this cannot be fully achieved by a single paper, and we do not intend to claim that our method *completes* this unification in a definitive or final sense, nor is it the *only* possible way to unify these fields. Rather, we see this work as:
> > > > > * a **concrete, principled step** toward such a unification via an AIF-based acquisition;
> > > > > * backed by **solid empirical evidence** on tasks where learning and optimization are tightly intertwined; and
> > > > > * a way to **invite more attention and follow-up work** in this direction.
> > > > >
> > > > > In this sense, we view our contribution as **foundational rather than final**: it introduces a unified objective where BO- and BED-like behaviors appear as special cases and shows that this perspective can be beneficial in hierarchical, preference-based settings.
> > > > >
> > > > > Indeed, the other two reviewers noted that the framework brings new insight ($\c$) and a generalized formulation ($\b$) that extends beyond existing BO/BED paradigms. This aligns with our goal: not to present the last word on unification, but to **open a coherent and principled path** that others (including us) can continue to build upon. We already have follow-up work in progress that extends this framework in specific directions, but we believe that a first step is to demonstrate that such a unifying perspective is both coherent and practically useful.
> > > > >
> > > > > We see this as the most valuable contribution a paper can make *at this stage* of the research effort:
> > > > > * to articulate the conceptual bridge,
> > > > > * demonstrate that it works, and
> > > > > * attract attention to a direction that has the potential to unify these communities over time.
> > > > >
> > > > > We also agree with you that the manuscript can better convey this message. In the latest revision, we made several changes aimed specifically at improving the presentation:
> > > > >
> > > > > **1. Reorganized the introduction (section 1) to sharpen the storyline**
> > > > >
> > > > > The introduction now explicitly:
> > > > > * motivates the “vacuum” between BO and BED for hybrid problems that require both *seeking knowledge* and *achieving goals*.
> > > > > * highlights how such problems span tasks with increasing epistemic and pragmatic complexity;
> > > > > * discusses how current approaches largely rely on problem-specific adaptations that do not generalize well across categories;
> > > > > * introduces our AIF-based viewpoint as a unified perspective; and
> > > > > * presents “pragmatic curiosity” as a general paradigm, followed by comprehensive empirical validation across three qualitatively different task families.
> > > > >
> > > > > **2. Clarified the scope of our claims in the Conclusion and Limitations (section 6)**
> > > > >
> > > > > We explicitly state there that the contribution is a **first, foundational step** toward unifying BO and BED, rather than a complete or final unification.
> > > > >
> > > > > **3. Expanded the related-work discussion (Appendix A)**
> > > > >
> > > > > We extended the related-work section in Appendix A to more clearly situate our framework within recent advances in Bayesian active learning and contextual / preference-based BO, and to sharpen how our perspective complements and differs from these lines of work.
> > > > >
> > > > > We hope these revisions help make the intent and scope of the paper clearer, and better communicate that our goal is to initiate a principled unifying direction rather than to claim that the full unification problem is already solved.

---

### Author Response · Authors · 2025-11-23
**Response to all reviewers**

$$
\def\a{\color{#648FFF}{\textsf{ZUrj}}} \def\b{\color{#E69F00}{\textsf{BqG3}}} \def\c{\color{#DC267F}{\textsf{xcCu}}}
$$

We thank all reviewers for their constructive and thoughtful comments. We greatly appreciate that **all reviewers** recognized both the **novelty** of our proposed framework and the **impressive experimental results**, which provide solid evidence for its effectiveness. We believe that our paradigm of *pragmatic curiosity* offers a generalized and systematized framework that covers a much wider range of problems than is typical for BO alone ($\b$), and brings new insights to the BED, BO, and AIF communities ($\c$).

The main concerns raised by the reviewers centered on: (i) insufficient clarification of experimental details ($\a$), (ii) positioning and comparison with the existing literature ($\a,\c$), and (iii) the under-explored effect of the curiosity parameters $\beta$ (and $\gamma$) ($\a,\c$). In the revised version, we added new experiments and carefully revised the introduction, related work, and experimental sections to address **all** of these points. All changes are marked in blue in the revised manuscript. The key updates are:

1. **Clarified and reorganized the experimental section.** ($\a$)
We moved key experimental details from the appendix into the main text and added a description of how the acquisition function is computed in practice using BoTorch.
2. **Clarified the relationship to classical BO and our baseline design.** ($\a,\c$)
   We explained why standard BO acquisition functions such as EI, UCB, or MES are not directly applicable to our non-canonical problem settings in Sec. 5.1 and Sec. 5.2, and described more clearly how we construct BO-type (optimization-focused) and BED-type (learning-focused) baselines tailored to each task.
3. **Added ablation studies on curiosity parameters.** ($\a,\c$)
   We introduced two new ablation studies analyzing how the curiosity parameters $\beta$ and $\gamma$ affect performance, and we provided practical design guidelines for choosing these parameters in different tasks.
4. **Extended the comparison to BOPE [1].** ($\a$)
   We added new experiments in Sec. 5.3 that directly compare our method with the baselines proposed in [1] (“Preference Exploration for Efficient Bayesian Optimization with Multiple Outcomes,” AISTATS 2022).
5. **Added a dedicated related-work subsection.**  ($\a$)
   We introduced Sec. 2.3 to discuss prior work on the synergy between BO and active learning/BED and to clarify how our contribution differs from existing approaches.

We hope that these additions and clarifications fully address the reviewers’ concerns and further highlight the novelty, generality, and practical value of our active-inference–based framework.

---

### Author Response · Authors · 2025-12-03
**Summary of Discussion**

$$
\def\a{\color{#648FFF}{\textsf{ZUrj}}} \def\b{\color{#E69F00}{\textsf{BqG3}}} \def\c{\color{#DC267F}{\textsf{xcCu}}}
$$

Dear AC,

Thank you again for managing the review process. Below, we provide a concise summary of the reviews, the discussion, and the revisions made.

In summary, we have very strong feedback: All reviewers highlighted the **novelty** of the framework and the **strength of the experimental results**, and agreed that the problem we address is important. Two reviewers gave scores of **8** and **6**. The third reviewer ($\a$) increased their score from **2 → 4** after our first rebuttal, and the only remaining hesitation is explicitly described as *subjective* and pertains to presentation and how strongly the “unifying BO and BED” message comes across, rather than to correctness or empirical support. We have further addressed this concern about presentation in our second round of response (which the reviewer can no longer comment on).

After two rounds of revisions and clarifications, **all of the concerns and questions** have been addressed. We summarize the strengths identified by the reviewers and our discussions below.

## Summary of strengths
Across the three reviews, the following strengths were emphasized:

- The paper introduces a **novel conceptual framework** based on active inference (“pragmatic curiosity”) that:
  - provides a **generalized and systematized formulation** covering a broader class of hybrid learning–optimization problems than typical BO alone ($\b$),
  - and brings **new insights** to the BO, BED, and AIF communities ($\c$).

- **All reviewers** recognized that the experimental section is **strong and comprehensive**, with:
  - diverse tasks where learning and optimization are tightly coupled,
  - solid empirical performance compared to BO/BED baselines,
  - and additional baselines and ablations added in response to reviewer feedback.

- **All reviewers** found the direction **promising and impactful**, especially the idea of treating explore–exploit as a unified inference problem rather than as separate heuristic stages.

## Summary of discussion

The main issues raised in the first round were:

**1. Clarity of experimental details** ($\a$)
**2. Positioning and comparison with existing literature** ($\a,\c$)
**3. Under-explored role of curiosity parameters $\beta$, $\gamma$** ($\a,\c$)

In the revised manuscript, we:

- clarified and reorganized the **experimental section** (Sec. 5), moving key details (dimensionality, baselines, computation) from the appendix into the main text, and
- added **new ablation studies** (Secs. 5.1, 5.3) and an **extended baseline comparison** (Sec. 5.3, Appendix C.4.4) to systematically address questions about baselines and parameter sensitivity.

Reviewer $\c$ explicitly acknowledged that these changes address their concerns and improve the presentation.
Reviewer $\a$ stated that the additional experiments and clarifications addressed “several” major concerns and justified raising the score, but still felt that the manuscript would benefit from further polishing of the exposition and the way the “unification” message is conveyed.

We took this seriously and, in the second revision, focused on sharpening the **framing and positioning**:

- We now **explicitly state** in the paper (Conclusion and Limitation, section 6) that our contribution is **foundational rather than final**:
  - we do **not** claim to *completely* solve the unification of BO and BED, nor to be the only possible unifying approach;
  - instead, we position the work as a **concrete, principled step** toward unification via an AIF-based acquisition, supported by strong empirical evidence, and intended to open a coherent path for follow-up work.

- We **reorganized the introduction** to make the storyline clearer:
  - identify the “vacuum” between BO and BED for hybrid problems that require simultaneously *seeking knowledge* and *achieving goals*;
  - show how existing methods handle these mostly via problem-specific heuristics;
  - introduce our AIF-based perspective as a unified view; and
  - present “pragmatic curiosity” as a general paradigm validated across three qualitatively different task families.

- We **expanded and clarified related work** (Appendix A) to better position our method within Bayesian active learning and contextual/preference-based BO.

Taken together, we believe the remaining concern from $\a$ about presentation has been fully addressed after this further revision.

---

> ### Author Response · Authors · 2025-12-03
> **Summary of key changes in the revision**
>
> After two rounds of discussion, the main changes are:
>
> 1. **Introduction (Sec. 1)** and **Conclusion & Limitation (Sec. 6).**
>    - Reorganized the introduction to sharpen the storyline by offering a stronger *placement* of the paper against prior works.
>    - Clarified in the Conclusion and Limitation section that we do *not* claim a complete unification, but a *foundational* step toward unifying BO/BED-style objectives.
>
> 2. **Related work (Appendix A).**
>    - Added a dedicated related-work section to more clearly position our framework within the BO, BED, and Bayesian active learning literature, including recent contextual and preference-based methods.
>
> 3. **Experiments (Sec. 5).**
>    - Moved key task and implementation details (e.g., dimensionality, baselines, acquisition computation via BoTorch) from the appendix into the main text to improve transparency and readability.
>    - Clarified the distinction between BO-type (optimization-focused) and BED-type (information-focused) baselines and how they are tailored for each non-canonical task.
>
> 4. **Curiosity parameter ablations (Secs. 5.1, 5.3).**
>    - Added ablation studies over $\beta$ and $\gamma$ and distilled practical guidelines for choosing them, demonstrating that performance is robust within reasonable ranges and that moderate “curiosity” improves performance.
>
> 5. **Extended comparison with BOPE [1] (Sec. 5.3, Appendix C.4.4).**
>    - Added experiments directly comparing to BOPE [1] in the composite preference-learning setting, showing that BOPE’s best-performing configuration is a special case of our nested objective and that our unified acquisition discovers higher-preference regions and is less sensitive to stage-wise design choices.
>
> [1] “Preference Exploration for Efficient Bayesian Optimization with Multiple Outcomes,” AISTATS 2022.
>
> ---
>
> In light of these revisions, together with the strong scores from the two reviewers (**8** and **6**) and a clear upward revision from $\a$ (**2 → 4, with an intention to increase if possible**), we believe all concerns have been satisfactorily addressed, and the paper is now in a strong position for acceptance.

---

### Meta-Review · Area_Chair_Xu2d · 2025-12-08

**Summary:**

This paper introduce "pragmatic curiosity", a new paradigm that resolves the classic explore-exploit dilemma by treating goal-seeking and information-seeking as two facets of a single, principled objective.

### Pros

* The idea of unifying goal-oriented and information-oriented tasks under one theoretical roof is conceptually appealing and motivated.

* The authors conducted extensive experiments, including complex composite tasks.

### Cons

* The primary claim of "unification" feels more like a narrative framing than a rigorous theoretical breakthrough. As noted by Reviewer ZUrj, the paper struggles to justify this unification beyond a high-level story, making the contribution feel somewhat hollow.

* The method introduces additional complexity and sensitive hyperparameters ($\beta, \gamma$) compared to standard BO/BED methods. While the authors provided guidelines, this adds a layer of fragility to the framework.

* Calculating EFE is inherently computationally expensive, as Reviewer BqG3 pointed out, and may become a bottleneck in large-scale or real-time scenarios.

### AC's evaluation

1. from reviews and rebuttals

This paper receives 862. Reviewer BqG3 (8) is very enthusiastic but provided a brief review with less technical scrutiny. Reviewer xcCu (6) is positive, focusing on approximations and parameters, and was satisfied by the clarifications. Crucially, Reviewer ZUrj (Initial Score 2) provided a detailed critique, questioning the experimental setup, missing baselines, and the "unification" story. After the rebuttal, ZUrj acknowledged the improvements (baselines added) but explicitly stated: "I do not think the paper is yet at the level of a weak accept (6), but if there were a '5' rating option, I would choose it." That means ZUrj still tends to reject this paper.

2. from AC's reading

I align with Reviewer ZUrj’s assessment and recommend Rejection. First, I am discounting BqG3's score (8) due to the lack of depth in their review. Second, the paper's main contribution relies heavily on its narrative of "unification." However, upon closer inspection, the method appears to be a specific AIF-based algorithm that mimics BO or BED depending on tuning, rather than a fundamental unification. While the authors' rebuttal work was impressive (which saved the paper from a strong reject), the gap between the paper's pitch and its actual technical contribution remains. It feels like a paper that needs a narrative overhaul to be accepted at ICLR.

**Reviewer Concerns:**

Resolved Concerns:

1. Missing Baselines (Reviewer ZUrj): The reviewer noted a lack of comparison with methods like BOPE. The authors added these comparisons in the rebuttal, demonstrating the method's effectiveness.

2. Parameter Sensitivity (Reviewer xcCu): Concerns about the difficulty of tuning $\beta$ and $\gamma$ were addressed with new ablation studies and guidelines.

3. Approximation Validity (Reviewer xcCu): Questions regarding the mathematical approximations in the EFE derivation were answered satisfactorily.

Outstanding Concerns:

1., The "Unification" Narrative (Reviewer ZUrj): This is the deal-breaker. Reviewer ZUrj argues that the paper forces a "unified framework" story onto a method that doesn't inherently solve the dichotomy between BO and BED. Despite the rebuttal, the reviewer remained unconvinced that the paper meets the bar for acceptance in terms of presentation and conceptual soundness.

2. Computational Cost (Reviewer BqG3): Even the most positive reviewer noted that calculating EFE is computationally expensive, limiting the method's scalability compared to standard approaches.

**Reviewer Scores:**

One person participated the discussion.

1. Reviewer BqG3 (8) will likely maintain 8. They are a fan of the work, but their review carries less weight due to its brevity and depth.

2. Reviewer xcCu (6) will maintain 6. They are in a "weak accept" state, satisfied but not championing the paper.

3. Reviewer ZUrj (2 -> 4): Started at 2. After the rebuttal, they moved to a hypothetical 4 or 5, explicitly stating it is not a 6.

---

### Decision · Program_Chairs · 2026-01-26

Reject